# WASp modulates RPA function on single-stranded DNA in response to replication stress and DNA damage

Seong-Su Han[1,2,6], Kuo-Kuang Wen[1,2,6], María L. García-Rubio [3], Marc S. Wold [4], Andrés Aguilera [3], Wojciech Niedzwiedz [5✉] & Yatin M. Vyas [1,2✉]

Perturbation in the replication-stress response (RSR) and DNA-damage response (DDR) causes genomic instability. Genomic instability occurs in Wiskott-Aldrich syndrome (WAS), a primary immunodeficiency disorder, yet the mechanism remains largely uncharacterized. Replication protein A (RPA), a single-strand DNA (ssDNA) binding protein, has key roles in the RSR and DDR. Here we show that human WAS-protein (WASp) modulates RPA functions at perturbed replication forks (RFs). Following genotoxic insult, WASp accumulates at RFs, associates with RPA, and promotes RPA:ssDNA complexation. WASp deficiency in human lymphocytes destabilizes RPA:ssDNA-complexes, impairs accumulation of RPA, ATR, ETAA1, and TOPBP1 at genotoxin-perturbed RFs, decreases CHK1 activation, and provokes global RF dysfunction. *las17* (yeast *WAS*-homolog)-deficient *S. cerevisiae* also show decreased ScRPA accumulation at perturbed RFs, impaired DNA recombination, and increased frequency of DNA double-strand break (DSB)-induced single-strand annealing (SSA). Consequently, WASp (or Las17)-deficient cells show increased frequency of DSBs upon genotoxic insult. Our study reveals an evolutionarily conserved, essential role of WASp in the DNA stress-resolution pathway, such that WASp deficiency provokes RPA dysfunction-coupled genomic instability.

[1] Department of Pediatrics, PennState College of Medicine, PennState Health Children's Hospital, Hershey, PA 17033, USA. [2] Division of Pediatric Hematology-Oncology, University of Iowa Stead Family Children's Hospital, Iowa City, IA 52242, USA. [3] Andalusian Center of Molecular Biology and Regenerative Medicine CABIMER, Department of Genome Biology, University of Seville-CSIC-University Pablo de Olavide, Seville, Spain. [4] Department of Biochemistry and Molecular Biology, University of Iowa Carver College of Medicine, Iowa City, IA, USA. [5] The Institute of Cancer Research, 237 Fulham Road, London SW3 6JB, UK. [6]These authors contributed equally: Seong-Su Han, Kuo-Kuang Wen. ✉email: wojciech.niedzwiedz@icr.ac.uk; yvyas@pennstatehealth.psu.edu

Wiskott-Aldrich syndrome (WAS) is an inborn error of immunity (IEI) caused by deficiency of WASp[1]. WAS lymphocytes show some combination of actin-cytoskeletal defect, gene transcription defect, and genome instability[2–8], manifesting clinically in immunodeficiency, atopy/autoimmunity, and lymphoid malignancy[9]. WASp supports ARP2/3-mediated actin-polymerization in the cytoplasm and RNA Pol II-mediated transcription in the nucleus[3,10]. Recent evidence links WASp to preserving genome integrity by modulating the cellular load of DSBs in human T and B lymphocytes, cells critical for adaptive immunity. WASp does so by suppressing the ectopic accumulation of pathologic R loops (formed by DNA-RNA hybrid and displaced ssDNA) that cause DSBs[11], and by enabling the early step of transporting DSB-ends for repair by the homology-directed repair (HDR) pathway[12]. Therefore, WASp-deficient lymphocytes manifest both the afferent arm (increased DSBs) and efferent arm (decreased DSB-repair efficiency) of the genome instability circuit. Furthermore, WASp deficiency undermines the nucleus-to-Golgi signaling elicited by DNA damage that is essential for cell survival after genotoxin-induced damage[13]. Altogether, these defects render WASp-deficient cells ill-equipped to manage genomic stress, endogenous or exogeneous. Thus, the collective evidence has greatly expanded WASp role from a cytoplasmic regulator of actin-cytoskeleton to a nuclear regulator of genome integrity.

A key factor tasked to preserve genome integrity is Replication protein A (RPA), a heterotrimeric protein containing RPA1, RPA2, RPA3 subunits, involved in DNA repair, replication, and recombination[14]. RPA accumulates at ssDNA sites generated from DNA-processing or stalled RFs. Although RPA is a high-affinity ssDNA-binding protein [Kd: $\sim 10^{-10}$ M], it is readily replaced by proteins that bind ssDNA with lower-affinity than RPA, during the temporal progression of the protein "handoff" reactions involved in fork protection and DNA repair[15]. This indicates that proteins and/or signals that can dynamically enable/disable RPA:ssDNA interaction must exist. As such, Saccharomyces cerevisiae (Sc) RPA employs Rtt105 (Protein-Regulator-of-Ty1-transposition-105) to optimize its binding with ssDNA[16,17]. Similarly, Xenopus RPA-interacting protein (XRIP) facilitates RPA nuclear localization[18]. These data suggest that RPA employs one or more "chaperons" to enable its interaction with ssDNA, and that this requirement is likely evolutionarily conserved. Yet, to date, how RPA-binding to ssDNA is modulated in humans is ill-understood. Here, we show that WASp is a critical factor that directly binds RPA and modulates its ssDNA-binding activity. Furthermore, we provide evidence that WASp:RPA alliance is required for maintaining genome integrity by influencing the RSR and the DDR in human lymphocytes and S. cerevisiae. This study establishes that RPA activity/function in higher and lower eukaryotes depends on WASp (or ScLas17) to efficiently manage DNA stress; and in the absence of WASp, RPA dysregulation triggers replication stress, DNA damage, and cellular/organismal dysfunction. The study, therefore, identifies WASp as an essential part of both normal DNA replication and DNA stress-resolution pathway, and unveils the sources of impaired RSR-linked genome instability in WASp-deficient lymphocytes.

## Results

### Genotoxins promote WASp accumulation at stressed RFs and its co-association with the markers of stressed DNA. To investigate a possible role of WASp in the RSR and DDR, we employed hydroxyurea (HU: 1 mM, 2 h; causes reduction of RF-velocity and RF-stalling) and camptothecin (CPT: 2 μM or 5 μM, 2 h; causes ssDNA nick, which after replication would become a single-ended DSB) as genotoxins. Using proximity ligation assay (PLA), we show in human T cells and B cells that these genotoxins induce WASp association in vivo with γH2A.X and RPA2(pSer33) (Fig. 1a), both known to co-accumulate at stressed RFs[19,20]. In contrast, WASp does not associate spontaneously (i.e., without damage) with these DNA stress markers. We next performed the quantitative in situ analysis of protein interactions at RFs (SIRF)[21], a technique that combines principles of iPOND (isolation of proteins at nascent DNA)[22] and PLA to visualize WASp enrichment at RFs at the single-cell resolution. We first verified in human T cells that PCNA (proliferating cell nuclear antigen) and GINS-CDC45 (replisome component) are constitutively present at unperturbed RFs, as SIRF positive controls (Fig. 1b). Like PCNA and CDC45, WASp is also constitutively present at unperturbed RFs, albeit at a lower frequency, in T and B cells (Fig. 1b), suggesting a role for WASp in normal DNA replication. However, following HU- or CPT-treatment, WASp enrichment at perturbed RFs is increased significantly in the WT T cells and B cells, reported by SIRF (Fig. 1b).

Because ssDNA stretches generated at stressed RF could be long, and thus some of the RPA on ssDNA may be far from EdU-labeled DNA, which could potentially underestimate SIRF foci data, we next verified the SIRF data by iPOND/Western blot. This showed that the enrichment of WASp and RPA2 at perturbed RFs is significantly increased over the steady-state levels following HU-induced RF perturbation, even at an early time-point of 30 min post-HU (Fig. 1c). Similarly, by Western blot we show that the relative abundance of WASp expression in the nucleus of human B cells also increases upon HU-stress relative to unstressed control (Supplementary Fig. S1). The collective findings suggest that WASp is part of the molecular apparatus that regulates both normal DNA replication and the RSR.

**WASp directly binds RPA.** Since RPA is a central component of the RSR and DDR apparatus, and because WASp and RPA co-accumulate at perturbed RFs (Fig. 1a-c), we asked if WASp and RPA interact directly. Using ELISA-based protein-protein binding assays, we show that native purified human WASp directly binds native purified heterotrimeric human RPA protein (RPA1, 2, 3; aka, RPA70, RPA32, RPA14) in vitro, in a dose-dependent manner (Fig. 2a). WASp binds in vitro to both the RPA heterotrimer-ssDNA mixture and RPA heterotrimer alone (Fig. 2a), but does not bind to ssDNA, dsDNA, or ssRNA alone (Supplementary Fig. S2a), denoting that WASp and RPA can interact directly. Moreover, purified human WASp does not bind purified human MAX (Myc-associated factor X), Bovine serum albumin (BSA), or Saccharomyces ScRpa protein (Fig. 2a), confirming specificity of WASp:RPA physical interaction. These in vitro findings align with the in-situ proximity of WASp and RPA2 (likely <10 nm distance from each other) in intact T and B cells by PLA, following HU- or CPT-induced DNA stress (Fig. 1a). Together, these results demonstrate that WASp is an RPA-interacting protein, in vitro and in vivo.

**WASp:RPA interaction is mediated by RPA1-binding motif in WASp.** We next sought to identify the WASp-domain(s) involved in RPA interaction. Because RPA1-interacting proteins express a consensus motif: D-φ-x-φ-D-D-φ-x-D-D (D: Aspartic acid; φ: hydrophobic or charged side-chain amino acid; x: any amino acid)[23,24] (Fig. 2b), we inspected the primary sequence of human WASp and found a putative RPA1-binding motif (RBM1) ($^{493}$D-E-D-E-D-D-E-W-D-D$^{502}$), located in the 3'-end acidic (A)-region of WASp's VCA-domain, which we show is evolutionarily-conserved down to yeast, and resembles the RBM1-consensus in ATRIP, ETAA1, and RAD9A, other known

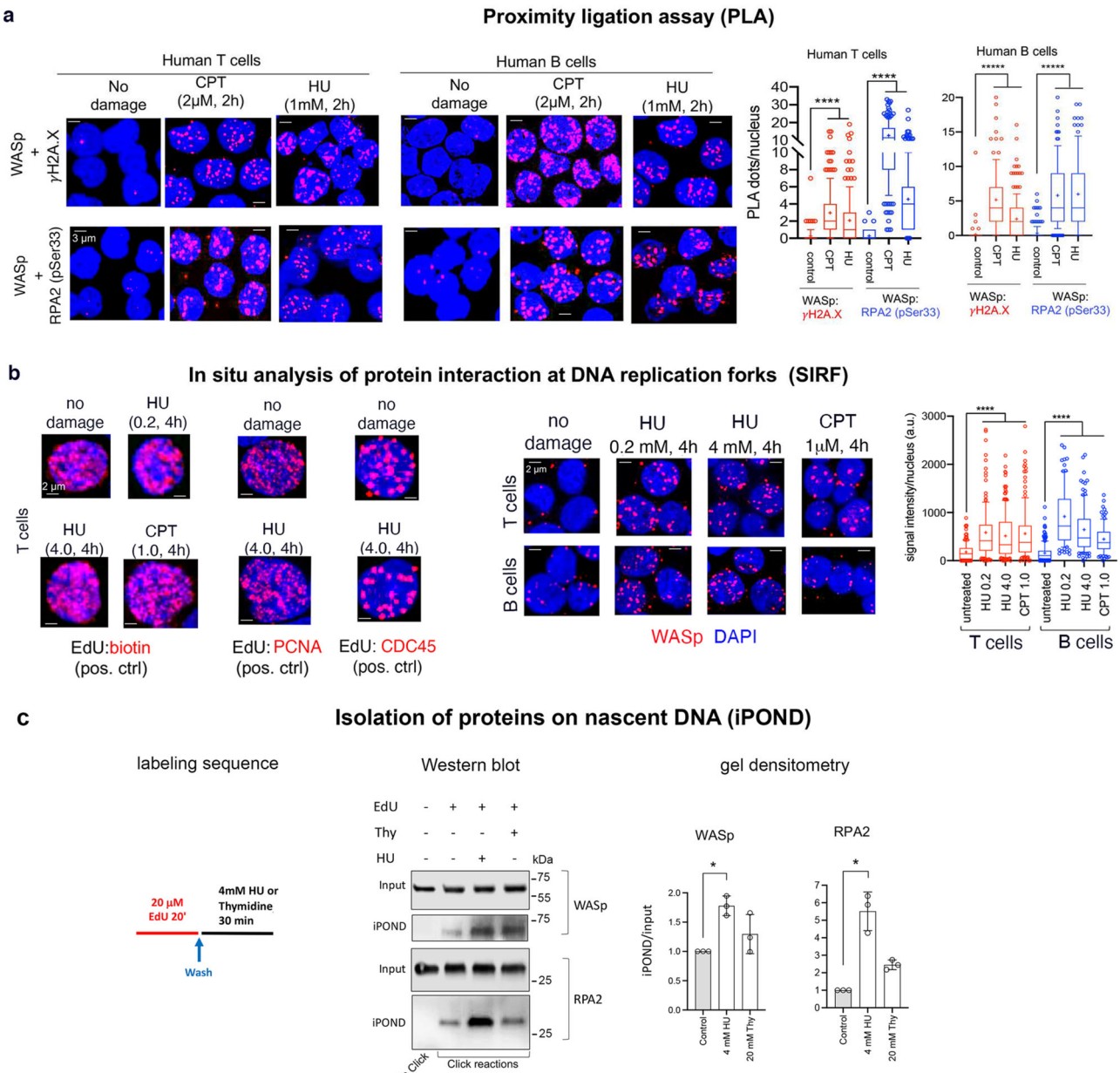

**Fig. 1 WASp is a DNA stress-response protein. a** Proximity ligation assay (PLA). Confocal immunofluorescence (IF) microscopy showing a representative set of collapsed composite IF images from a z-stack of ~30-40 images acquired per optical field. Shown are the PLA signals between the indicated interacting proteins induced by the indicated genotoxins or no damage control in human T and B cells, along with their corresponding PLA statistics in box-and-whisker plots (whiskers @10–90%, horizontal bar denotes median, "+" denotes mean). In both panels, unpaired, two-tailed, Mann-Whitney nonparametric $p$ values ****<0.0001, $n = 150$ cells/condition. Scale bar: 3 μm. **b** In situ analysis of protein interactions at replication forks (SIRF). *Right*, Representative z-stack collapsed composite confocal IF images showing SIRF signals between 5-ethynyl-2'-deoxyuridine (EdU)-labeled nascent DNA and the indicated protein in steady-state (no damage) or after HU- or CPT-induced DNA perturbation, along with their corresponding statistics ($n = 100$ cells). *Left*, representative SIRF of positive and negative controls (ctrl), under replicative stress or no stress/damage. Statistical details as per **a**. Scale bar: 2 μm. **c** Isolation of proteins on nascent DNA (iPOND). *Left*, schematic of nascent DNA labeling, in which red line denotes DNA labelled with EdU, followed by black line denoting chase into media containing either hydroxyurea (replication stressor) or thymidine (control for true replication proteins that will not enrich in this sample due to EdU displacement). *Middle*, shows Western blots of input and iPOND purified proteins under indicated conditions, including no EdU sample (no-click negative control). *Right*, bar graphs depict gel densitometric quantitation of iPOND/Western blot signals normalized to their respective inputs. $n = 3$, +SEM. Mann-Whitney unpaired two-tailed nonparametric $p$-value *<0.01. Source data are provided as a Source Data file.

RPA1-binding proteins (Fig. 2b). To test the functionality of RBM1, we generated WASp-mutant lacking aa:493-502 (ΔRBM1*WASp) (Supplementary Fig. S3a). We purified WT*WASp and ΔRBM1*WASp proteins, and by ELISA method show that the in vitro binding of ΔRBM1*WASp to purified heterotrimeric-RPA is dramatically reduced relative to

WT*WASp (Fig. 2c). Notably, ΔRBM1*WASp mutant can still bind purified Arp2 protein (Fig. 2c), the latter previously shown to bind VCA-domain with higher affinity than Arp3[25], thus providing subdomain-delimited separation-of-function for actin-polymerization versus RPA-activity. Next, we expressed GFP-tagged ΔRBM1*WASp-mutant or WT*WASp into WASp-

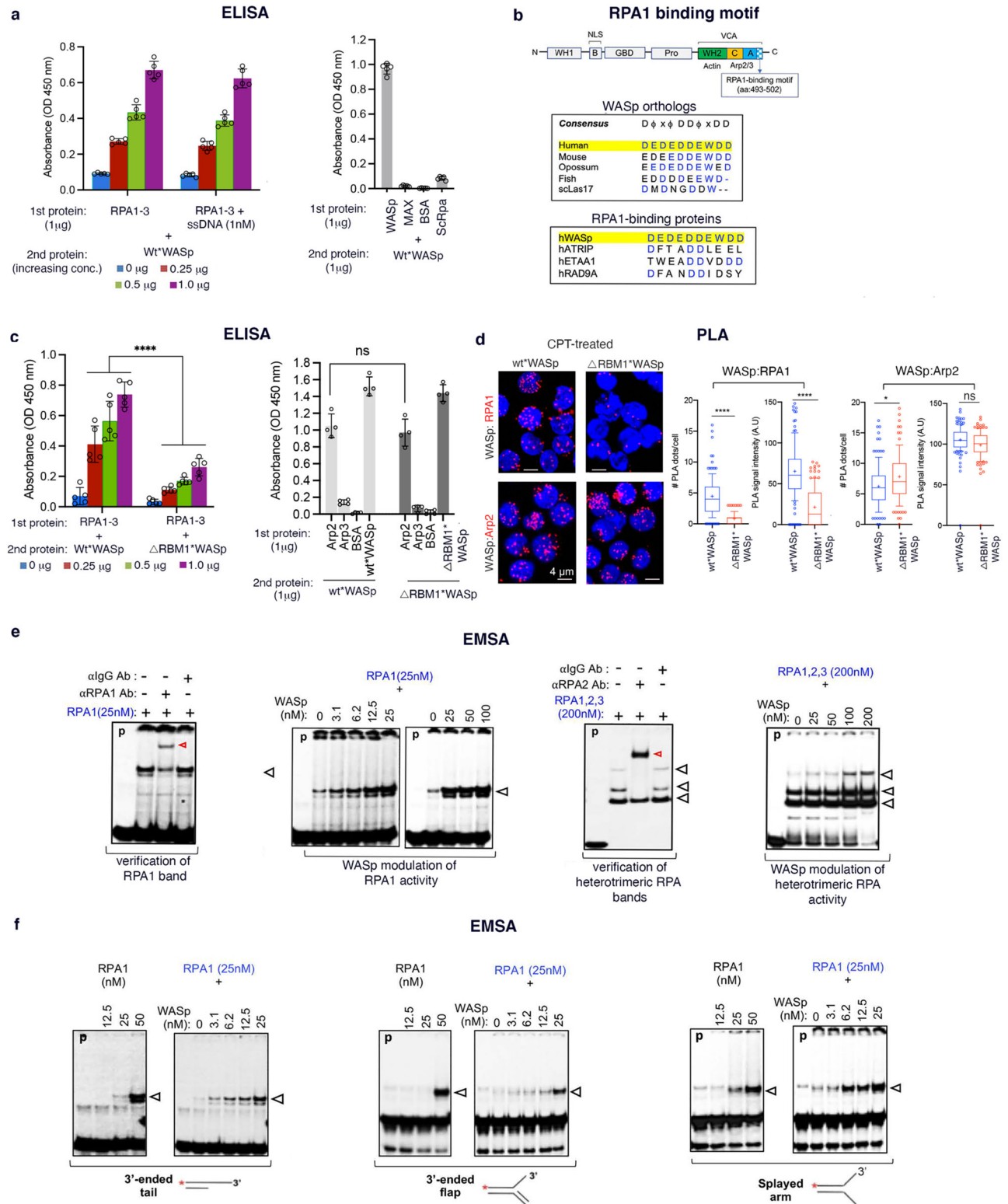

deficient (WKO) T cells (Supplementary Fig. S3b) and show by the PLA method in the FACS-enriched GFP-expressing T cells that the in vivo binding of transfected ΔRBM1*WASp with endogenous RPA1 is significantly lower relative to that of transfected WT*WASp after 2 h of CPT treatment (Fig. 2d). In contrast, transfected ΔRBM1*WASp can still bind endogenous Arp2 in the cytoplasm and nucleus (PLA signals captured in both subcompartments; cytoplasmic PLA signals appear to be concentrated in the region typically occupied by Golgi and/or

microtubule-organizing center) (Fig. 2d). Together, these results indicate that RBM1 primarily mediates WASp interaction with RPA, in vitro and in vivo.

**WASp enhances the ssDNA-binding activity of RPA**. To elucidate a functional relevance of the WASp:RPA association, we investigated how WASp modulates RPA's innate function of binding ssDNA. In EMSA-binding assays, we show that adding

**Fig. 2 WASp directly binds RPA and enables RPA binding to ssDNA. a**, ELISA. In vitro protein-protein interaction monitored by ELISA for the indicated purified proteins at the indicated concentrations. 1st protein is coated onto the plate at a fixed concentration; 2nd protein is added at the indicated increasing concentrations. hRPA1-3, human heterotrimeric complex of RPA1, 2, 3; hMAX, human Myc-associated factor X; BSA, bovine serum albumin; ScRpa, Saccharomyces Rpa; WT*WASp, wild-type WASp. The displayed data are mean+SEM, n = 5 independent assays. The intrinsic property of WASp molecules to spontaneously oligomerize (via VCA:VCA-domain interaction) served as +ve control, and is shown for the highest concentration (1 μg) of WT*WASp protein. **b** Schematic of the multi-modular domain structure of human WASp showing WH1-domain, Basic-domain containing the NLS (B), GTPase-binding domain (GBD), Polyproline-domain (Pro), VCA-domain (WH2, C, A subdomains), in which the location of evolutionarily conserved RPA1-binding motif (RBM1) within the VCA-domain is shown. WH2 (aka, V region) binds monomeric G-actin, C and A regions bind Arp2/3-complex. Amino-acid alignments of human WASp's RBM1 with those of other proteins and species are shown, in which residues that are conserved (in evolution) and common (with other RPA1-binding proteins) are highlighted in blue. The prototypical RBM1-consensus is shown at the top, D: Aspartic acid; ϕ: hydrophobic or charged side-chain amino acid; x: any amino acid. **c** ELISA. Shown is the binding efficiency of purified heterotrimeric RPA1-3 protein (on left) or purified Arp proteins (on right) with either purified Wt*WASp or RBM1-deleted WASp mutant (ΔRBM1*WASp) at the indicated concentrations. Physical interactions of WT*WASp:WT*WASp and mutant ΔRBM1*WASp:ΔRBM1*WASp (from spontaneous oligomerization of their respective VCA-domains) served as +ve controls. n = 3 independent assays, mean+SEM. p-value: ****<0.0001. ns, nonsignificant by Mann–Whitney unpaired two-tailed nonparametric. **d** Proximity ligation assay. Z-stack collapsed composite confocal IF images showing PLA signals (dots) between the indicated protein pairs after transfecting WASp-deficient human T cells with GFP-tagged Wt*WASp or ΔRBM1*WASp mutant after CPT-induced damage. The images/data are for the GFP+ cells enriched by FACS-sorting. Box plots are from n = 150 cells, Mann-Whitney unpaired two-tailed nonparametric p-value: ****<0.0001; * = 0.01; ns, nonsignificant. DAPI (in blue) demarcates the nucleus. Scale bar: 4 μm. EMSA. Assays performed using the indicated purified proteins at the indicated concentrations (nM) against fixed concentration of ssDNA (panel **e**) or other 3'-ended DNA structures (panel **f**). Data is representative of at least 3 replicates per condition. Red arrows indicate antibody super-shifted RPA band; other arrows indicate location of non-shifted RPA bands. Other related EMSA results are shown in Supplementary Fig. S2.

increasing concentrations of purified WASp to a fixed mixture of ssDNA oligo-probe (61-nt) plus either purified-RPA1 or heterotrimeric-RPA protein, results in an augmented RPA:ssDNA complexation (Fig. 2e). Maximum RPA1:ssDNA complexation is observed with equal molar concentrations of WASp, i.e., 25 nM (Fig. 2e). In contrast, addition of purified FANCA, XRCC2, DNMT1, MYC, NF-κB1, STAT1, or BSA, does not change the binding affinity of RPA1 to ssDNA (Supplementary Fig. S2b), denoting unique effect of WASp on RPA:ssDNA interaction. These results suggest that a direct interaction between WASp and RPA is modulating RPA interaction with ssDNA, either by increasing affinity or promoting the stability of the RPA:ssDNA-complex.

To further refine this finding, we tested WASp modulation of RPA1-binding to multiple other DNA conformations. Because RPA1 binds ssDNA with a preference for 3'-protrusions (3'-ended tail, 3'-ended flap, 3'-splayed arm) over 5'-protrusions[26,27], we first verified these RPA1-binding preferences by EMSA. We show that RPA1 binding to the 3'-ended ssDNA protrusions of 30-nt (considered both optimum and stable ssDNA-binding length for hRPA)[27] is also increased by WASp in a dose-dependent manner (Fig. 2f). Notably, WASp increases binding of RPA1 to DNA-structures containing ssDNA-protrusion even at a low concentration (25 nM RPA1), which otherwise show a modest binding on its own, i.e., without added WASp (Fig. 2f). These data suggest that WASp improves the efficiency of RPA1 binding to ssDNA. In contrast, WASp does not ectopically enable RPA binding to nucleic-acid structures that normally do not efficiently bind RPA, e.g., ssRNA or DNA Holliday junction[26,27](Supplementary Fig. S2c). Together, the data demonstrate specificity of WASp effect on the intrinsic activity of RPA to bind ssDNA.

Finally, since RPA-binding to ssDNA is dynamic, involving cycles of association-dissociation-reassociation of RPA-subunits to ssDNA[28–30], we tested if the effect of WASp on RPA involves modulating the DNA binding-activity of individual RPA-subunits. To this end, we used purified RPA-subunits containing DNA-binding domain (DBD) DBD-F/A/B (RPA1), DBD-A/B (RPA1), and DBD-D/wh/E (RPA2/RPA3), (Supplementary Fig. S2d), which we had previously generated and characterized[29–31]. In EMSA-binding assays we show that purified WASp increases the ssDNA-binding activity of RPA1-

subunits DBD-F/A/B and DBD-A/B (Supplementary Fig. S2e), considered high-affinity DNA-binding domains (DBDs)[29–31]. In contrast, WASp does not ectopically induce DNA-binding of RPA2/RPA3-subunits DBD-D/wh/E (Supplementary Fig. S2e), considered the trimerization core that intrinsically has low-affinity for binding ssDNA. This data suggests that WASp likely participates in optimizing the interaction of RPA's high-affinity DBDs with the available binding sites on ssDNA, i.e., by modulating RPA's conformational state that favors more OB-fold domains to bind ssDNA[32]. Together, our results indicate that WASp directly enhances the binding and/or stabilization of RPA to multiple ssDNA conformations.

**WASp-deficiency disrupts genotoxin-induced RPA:ssDNA-complex formation.** To investigate the role of WASp in RPA-dependent RSR and DDR, we tested how WASp-deficiency resulting from patient-derived mutation in B cells (WAS03) or CRISPR/Cas9-mediated WASp depletion in T cells (ND1-WKO) (Supplementary Fig. S2f, WASp-expression profile) influences RPA activity at perturbed RFs. By SIRF, we show that the enrichment of endogenous RPA2(pSer33) at HU- and CPT-perturbed RFs (4 h post-treatment) is significantly increased relative to unperturbed RFs in WT T cells (Fig. 3a), which is consistent with an essential role of RPA in the RSR and DDR signaling. In contrast, the enrichment of RPA2(pSer33) is significantly decreased in WKO T cells relative to WT control, this despite a significant increase in γH2A.X enrichment at perturbed RFs relative to unperturbed RFs in both WT and WKO T cells (Fig. 3a). Notably, expression of total cellular RPA1 protein as well as the nuclear localization of RPA1 in WKO T cells is comparable with WT T cells in the steady-state in vitro culture condition (Supplementary Figs. S2g, S2h), which rules out the possibility that the decreased occupancy of RPA on stressed RFs is due to an overall reduction in the amount of RPA present in WASp-deficient cells. Similarly, a significant reduction in the SIRF-enrichment of RPA at perturbed RFs is observed also in WAS03 B cells relative to normal B cells (Fig. 3a).

We next verified the SIRF data by iPOND/Western blot. This showed that the enrichment of endogenous RPA2 at HU-perturbed RFs is significantly lower in WKO T cells relative to WT (Fig. 3b). Furthermore, in EMSA-binding assays we show

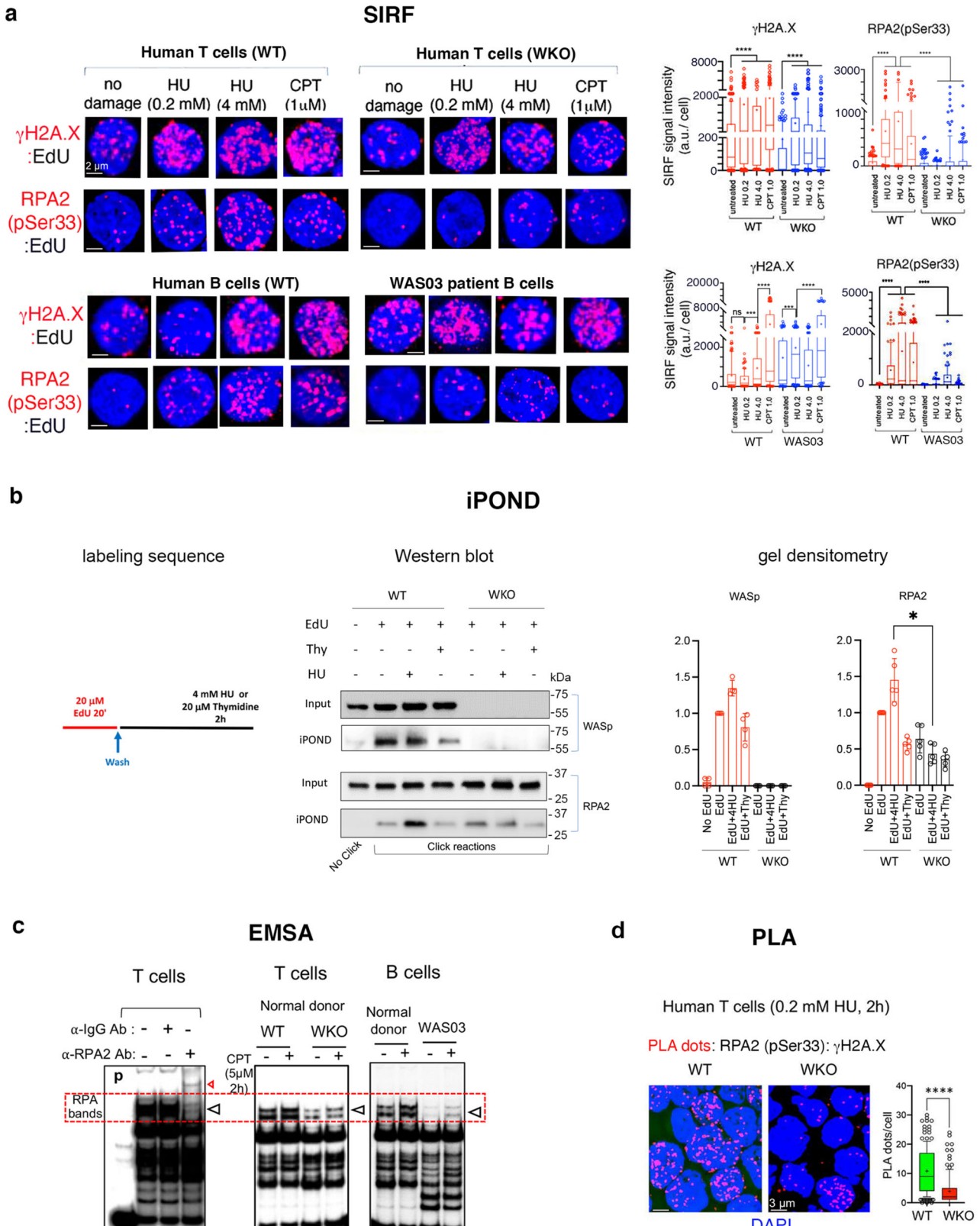

that the ssDNA-binding activity of endogenous heterotrimeric-RPA protein in the nuclear lysates of CPT-treated (5 μM; 2 h) WKO T-cells and patient-derived WAS03 B-cells is decreased relative to WT cells (Fig. 3c). Finally, we show by PLA that the co-association of RPA2(pSer33) with γH2A.X in situ is also impaired in HU-treated WASp-deficient T-cells compared to WT T-cells (Fig. 3d). Together, the data suggest that the targeting and/or

binding of endogenous RPA to the sites of DNA-damage or replication-stress is compromised in the absence of WASp.

**WASp-deficiency impairs activation of ATR/CHK1-signaling by disrupting TOPBP1 and ETAA1 enrichment at perturbed RFs.** Since ATR recruitment to RPA-coated ssDNA is required

**Fig. 3 WASp-deficiency impairs RPA occupancy at perturbed RFs. a**, SIRF. Shown are the representative z-stack collapsed composite confocal IF images and their box-and-whisker plots quantifying the enrichment of the indicated proteins at RFs, unperturbed or perturbed by the indicated genotoxins (post-4h treatment) in human T and B cells. $n = 150$ cells, Mann-Whitney unpaired two-tailed nonparametric $p$-value: **** <0.0001. WKO T cells and WAS03 B cells are WASp-deficient (see Supplementary Fig. S2f for WASp expression profiles). a.u. denotes arbitrary units. Scale bar: 2µm. **b**, iPOND. *Left* panel, schematic of nascent DNA labeling, description as per the legend for Fig. 1c. *Middle* panel, shows Western blots of input and iPOND purified proteins under indicated conditions for wild-type (WT) and *WAS* knock-out (WKO) isogenic pair of human T cells. iPOND signal for WASp in WKO T cells served as a negative control. *Right* panel, bar graphs depict gel densitometric quantitation of iPOND/Western blot band signals normalized to their respective inputs. $n = 3$, +SEM. Mann-Whitney unpaired two-tailed nonparametric *<0.01. Source data are provided as a Source Data file. **c**, EMSA/super-shift assays. Experiments performed using purified nuclear extracts from the indicated cell types, treated (+) or not (-) with CPT at indicated dose/duration. The location of endogenous heterotrimeric RPA bands, as verified by anti-RPA2 antibody mediated super-shifted band (left panel, red arrow denotes super-shifted band). Red hatched box denotes the general location of the endogenous RPA1-3 bands, verified by supershift. p, probe only lane. Data is representative of $n = 3$ independent assays. **d**, PLA. Representative z-stack collapsed composite confocal IF images and their quantification showing RPA2 localization at HU-mediated replication-stress sites (monitored by γH2A.X) post-2h in T cells, wild-type (WT) and WASp-deficient (WKO). Box-and-whisker plots, $n = 150$ cells, Mann-Whitney unpaired two-tailed nonparametric $p$-value: **** <0.0001. Scale bar: 3µm.

---

for the upregulation of ATR-kinase activity and phosphorylation of its substrate CHK1[33,34], we tested if the observed RPA:ssDNA association defect is sufficient to subvert ATR signaling at perturbed RFs in WASp-deficient cells. By SIRF, we show that ATR enrichment is increased at genotoxin-perturbed RFs compared to unperturbed RFs, in both WT (normal) T cells and B cells (Fig. 4a). In contrast, ATR enrichment at perturbed RFs is significantly decreased in WASp-deficient T and B cells (Fig. 4a). Moreover, enrichment of both ETAA1 and TOPBP1, the 2 canonical ATR-kinase activators, at perturbed RFs is also significantly reduced in WASp-deficient T cells relative to WT (Fig. 4a).

To directly test how WASp influences ATR:ETAA1 and ATR:TOPBP1 co-associations under replication-stress or DNA-damage, we performed PLA experiments. These show that HU (low or high dose)-induced co-association (reported by in situ proximity) of ATR with both ETAA1 and TOPBP1 was significantly increased in WT T cells relative to WASp-deficient T cells. Under CPT-condition, however, ATR:TOPBP1 co-association was somewhat increased in WKO T cells relative to WT T cells, whereas that of ATR:ETAA1 was comparable (Fig. 4b).

Finally, since dimerization of the TOPBP1 and ETAA1 oligomeric-complexes is required for optimal ATR function[35,36], we next tested how RF-resident defects of both TOPBP1 and ETAA1 influences ATR function of activating CHK1. Western blot analyses show that although genotoxins increase CHK1(pSer345) levels in both WT and WKO T cells, the magnitude of this increase normalized to total CHK1 is significantly lower in WKO T cells relative to WT T cells, this despite a comparable (or even higher) degree of DNA damage/replicative-stress (reported by γH2A.X) (Fig. 4c). CHK1(pSer345) activation defect was also observed in WAS patient-derived B cells (WAS03) subjected to HU-mediated replication stress (Fig. 4d). To further substantiate the necessity of RPA1:WASp interaction in the activation of ATR-mediated checkpoint, we show that *WAS*-null T cells re-expressing the WASp-mutant selectively lacking the RPA1-binding motif (ΔRBM1*WASp) also shows reduced CHK1(pSer345) activation relative to cells re-expressing WT*WASp control (Fig. 4e). We conclude that disruption of RPA:ssDNA complexation in WASp-deficient cells impairs ATR-ETAA1-TOPBP1 signaling and global checkpoint activation during the RSR and DDR.

**WASp-deficiency causes global RF dysfunction during replicative stress.** To understand how WASp deficiency-linked RPA-ATR-CHK1 signaling defect sensitizes T and B cells to replication stress, we conducted DNA fiber assays by labelling cells with sequential pulses of iododeoxyuridine (IdU) and chlorodeoxyuridine (CldU) in the presence or absence of HU to determine RF events (Fig. 5a-d).

First, unperturbed (no added genotoxins) WT and WASp-deficient T cells showed similar RF progression speed, with an average RF-velocity of ~0.6 kb/min for both WT and WASp-deficient (WKO) T cells (Fig. 5a). In contrast, HU-perturbed WKO T cells showed a significantly lower mean RF-velocity of ~0.02 kb/min compared to WT T cells, which showed mean RF-velocity of ~0.06 kb/min (Fig. 5b). WASp-deficient B cells (WAS03) also showed increased RF stalling relative to WT B cells following HU treatment (Supplementary Fig. S4).

Second, WASp-deficiency also affects replication resumption after release from HU, as monitored by RF restart (Fig. 5c). In WT T cells, ~85% of RFs have restarted at 30 mins post-HU removal (i.e., 2nd CIdU-track display at least 50% length relative to 1st IdU-track) with only 15% failing to restart (i.e., absent 2nd CIdU-track) (Fig. 5c). By contrast, in WKO T cells, ~60% of RFs fail to restart (i.e., 1st IdU-tracks only, or IdU-track followed by a dot-like 2nd CIdU signal) (Fig. 5c). Impaired replication resumption was seen also in WAS03 B cells after lifting HU stress (Supplementary Fig. S4).

Third, since impaired RF restart could occur from degradation of stalled RFs and ATR defect is known to provoke RF degradation[37,38], we tested if WASp is involved in protecting nascent DNA from degradation. To this end, we sequentially-labeled DNA with IdU followed by CIdU for 25 min each, and then induced replication stress with 4 mM HU for 4 h. We show that in those RFs that were dual-labelled (sequentially, but not overlapping, green and red tracks), the ratio of CIdU/IdU track lengths was ~1.5 for WT T cells, while in WKO T cells, the ratio was ~0.5, indicating that CIdU track-length was significantly shorter after HU-treatment, consistent with an increased degradation of HU-stalled RFs (Fig. 5d).

Together, these results implicate WASp in influencing global replication program under replicative stress and propose that increased degradation of stalled RFs contribute in part to genomic instability in WASp-deficient cells.

**WASp-deficiency causes heightened genome-instability in genotoxin-treated human T and B cells.** Next, we analyzed the effect of fork dysfunction on the cellular DSB load as a measure of genome instability. Employing neutral comet assays (monitors DSBs), we show that the frequency of DSBs, as inferred from comet-tail moments, is significantly higher in WASp-deficient T cells relative to WT, under genotoxic stress (Fig. 5e). As such, we observe a higher frequency of comet tail moments in WKO T cells relative to WT, even in the unstressed cells (Fig. 5e), suggesting spontaneous genome instability likely arising from the accumulation of pathological R-loops and/or impaired HDR in

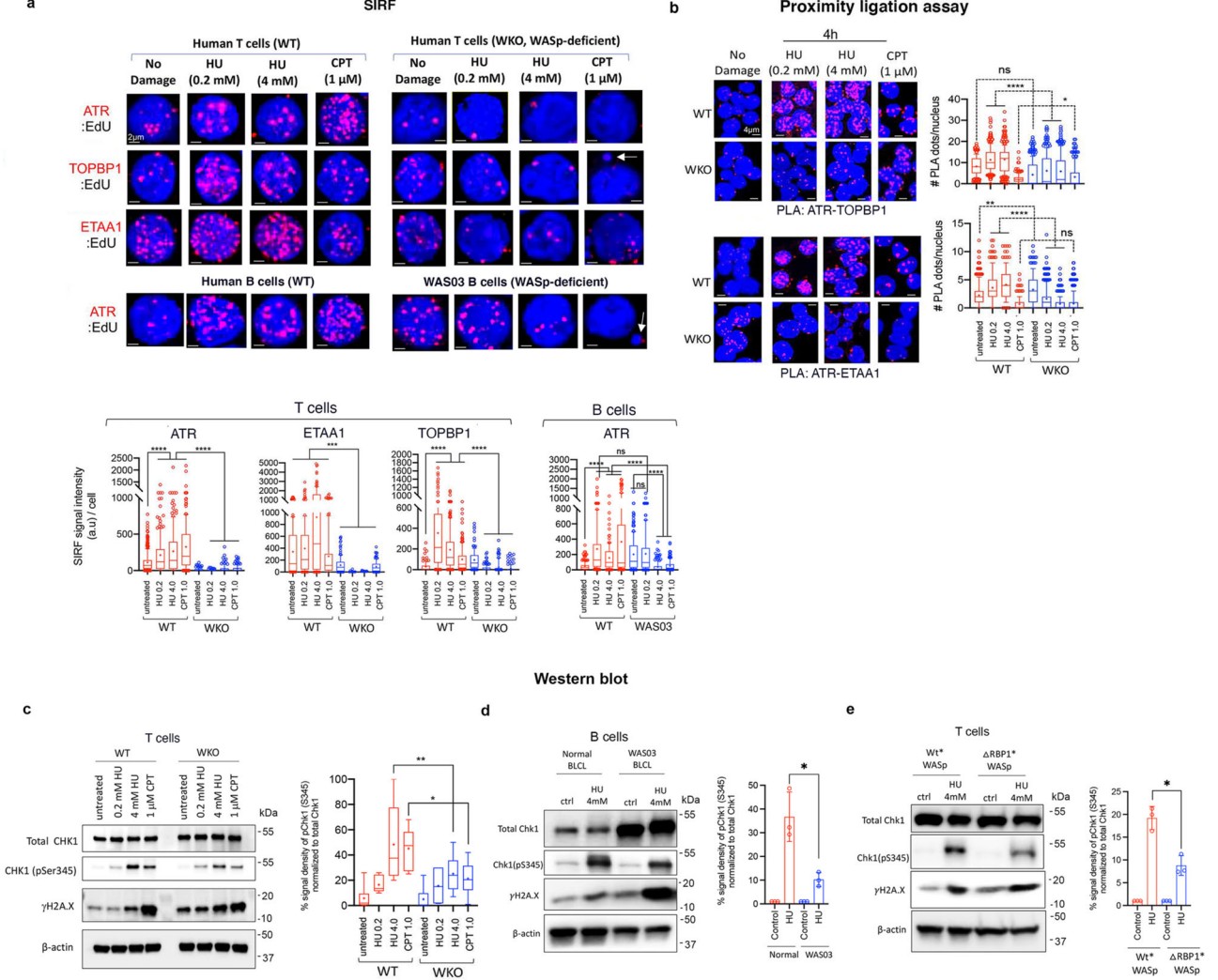

**Fig. 4 WASp deficiency disrupts ATR signaling at perturbed RFs and impairs global CHK1 activation. a**, SIRF. Shown are the representative z-stack collapsed composite confocal IF images and their box-and-whisker plots quantifying the enrichment of the indicated proteins at RFs, unperturbed or perturbed by the indicated genotoxins and doses (post-4h treatment) in human T and B cells, WT or WASp-deficient (WKO T cells; WAS03 B cells). In box-and-whisker plots, whiskers @10-90%, horizontal bar denotes median, "+" denotes mean. The box-and-whisker plots are from $n = 150$ cells analyzed, Mann-Whitney unpaired two-tailed nonparametric $p$-value: **** <0.0001; *** <0.001; ns, nonsignificant. a.u. denotes arbitrary units. Arrows show WASp-deficient T and B cells with micronuclei formation (See Fig. 5f for additional data on micronuclei). Scale bar: 2μm. **b**, PLA. Representative z-stack collapsed composite confocal IF images and their quantification plots for the indicated protein:protein interactions in T cells, wild-type (WT) and WASp-deficient (WKO), treated with the indicated genotoxin (post-4h treatment), or no treatment. In box-and-whisker plots, whiskers @10-90%, horizontal bar denotes median, "+" denotes mean. The box-and-whisker plots are from $n = 150$ cells analyzed, Mann-Whitney unpaired two-tailed nonparametric $p$-value: **** <0.0001; ** <0.001; *<0.01; ns, nonsignificant. Scale bar: 4μm. **c-e**, Western blot. Representative images of the indicated proteins expressed in total cell extracts of human T cells, WT and WKO (panel c), B cell lines (Normal donor and WAS03 patient) (panel d), and WASp-deficient (WKO) human T cells stably-transfected to re-express GFP-tagged WT*WASp or RPA1-binding domain-deleted mutant of WASp (ΔRBP1*WASp) (panel e) treated with the indicated genotoxin for 4 h or untreated (control, ctrl), along with their gel densitometric analyses. In box-and-whisker plots, whiskers @10-90%, horizontal bar denotes median, "+" denotes mean. $n = 4$ independent assays. In box plots shown in panels d and e, data is for mean±SEM. $n = 3$ independent assays. Mann-Whitney unpaired two-tailed $p$-value: *<0.01, **<0.001. In panel c, the $p$-values for HU (0.2 low-dose) and untreated (no damage) comparing WT and WKO T cells were not significant. Source data are provided as a Source Data file.

WASp-deficient cells in the in vitro culture conditions[11,12]. Furthermore, employing confocal immunofluorescence imaging of DAPI-labelled cells, we show increased micronuclei formation, likely due to sequestered damaged DNA caused by unresolved stressed RFs, in WASp-deficient T and B cells compared to WT controls treated with CPT (Fig. 5f). Together, the data imply increased DNA damage in WASp-deficient cells.

Because cell-cycle phase influences DSB repair pathway choice (SSA vs. HDR vs. NHEJ), dependent partly on whether end-resection is activated (S/G2 phase) or not (G1 phase), we next

analyzed the effect of fork dysfunction on cell-cycle distribution profiles. Flow-cytometry analyses revealed that, like ETAA1-deficiency in human HCT116 and HeLa cells[24], WASp-deficiency also does not markedly alter the overall cell-cycle distribution in human T and B cells (Supplementary Fig. S5). Under low-dose HU, both WT and WASp-deficient T and B cells show intra-S checkpoint activation. Under high-dose HU and CPT, WT and WASp-deficient T and B cells show a combination of G1 arrest, intra-S, and/or G2/M checkpoint activation with some notable differences in WT and WKO T-cells. Specifically, under CPT-

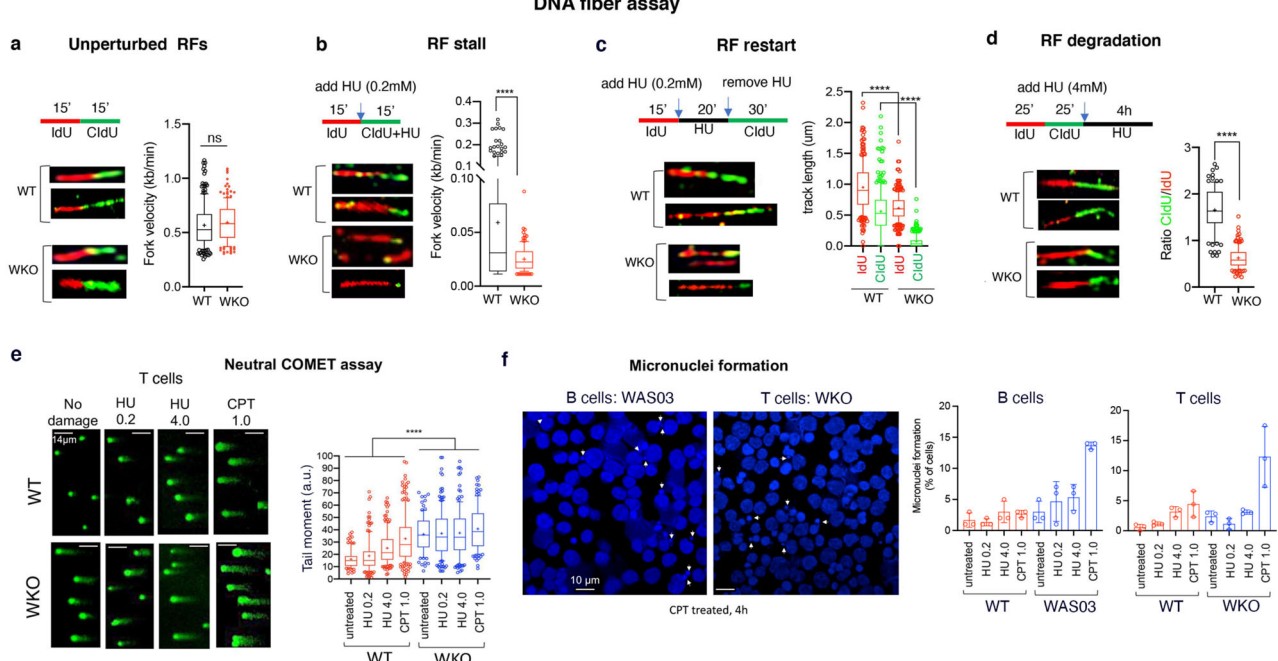

**Fig. 5 WASp-deficiency undermines RF integrity and causes genome instability. a–d,** DNA fiber assays showing 4 different labeling protocols, their representative replication track images, and their corresponding RF statistics in human T cells (WT, WKO isogenic pair) under unperturbed (panel a) or HU-perturbed (panels b–d) conditions. For assays in which the 1st and 2nd labeling times were similar, RF velocity was calculated (panels a, b), whereas, when labeling times were dissimilar, the individual RF track length was calculated (panel c). For quantifying RF degradation (panel d), the ratio of 2nd:1st labeled track lengths was calculated only in those tracks that showed sequential, but not overlapping, dual-labeling. The box-and-whisker plots (whiskers @10-90%, horizontal bar denotes median, "+" denotes mean) are from n = 150 tracks analyzed from 3 independent experiments. Mann-Whitney unpaired two-tailed nonparametric p-value: **** <0.0001; ns, nonsignificant. See Supplementary Fig. S4 for additional RF data in human B cells. **e,** Representative confocal IF images of the neutral comet assay reporting on the frequency of DSBs in human T cells (WT, WKO), untreated or treated with indicated genotoxins and doses for 4 h. The box-and-whisker plots (whiskers @10-90%, horizontal bar denotes median, "+" denotes mean) reporting on the tail-moment are from n = 150 cells analyzed, Mann-Whitney unpaired two-tailed nonparametric *p*-value: **** <0.0001. a.u. denotes arbitrary units. Scale bar: 14μm. **f,** *Left,* Representative confocal IF images of micronuclei formation (indicated by arrows) in DAPI-stained WASp-deficient T and B cells are shown for the CPT condition at 4 h. *Right,* statistical bar graph data, mean +SEM, *n* = 150 cells for all genotoxin conditions from n = 3 independent assays. Scale bar: 10 μm.

induced damage, WKO T cells showed a higher percentage of G1-arrested cells (~70% at 24 h; ~44% at 48 h) compared to WT T cells (~37% at 24 h; ~20% at 48 h). These cell-cycle profiles suggest that WASp likely influences DSB repair pathway choices in a context and cell-type dependent manner, as further evidence from our yeast studies described later. Together, our results indicate that WASp is important for maintaining genome stability in human lymphocytes.

**Yeast Las17-inactivation manifests RPA and recombinational DNA repair defects.** Since WASp has an ortholog in *Saccharomyces cerevisiae*, Las17, we did a genetic analysis to test whether results in human cells can be extrapolated to yeast to establish conservation of function in eukaryotes. Because in *S. cerevisiae*, homologous recombination is the primary mechanism for repairing DSBs, we used Las17-deficient *Saccharomyces* mutants carrying the *las17-14* allele (see *Methods* section) to directly test WASp (Las17) role in HDR. In addition, we generated a *las17* auxin-inducible degron strain (*las17-aid*) (see *Methods* section) to be able to conditionally deplete cells of Las17 expression after auxin (IAA) addition. By Western blot, we show that the degron allele effectively depleted the Las17 protein expression after IAA addition (Supplementary Fig. S6), thus validating the auxin-based degron system. Notably, hypersensitivity to DNA-damaging agents HU and Methyl methanesulfonate (MMS) is seen in both the *las17-14* and the *las17-aid* mutants, the latter in the

presence of IAA that activates the degron (Fig. 6a). This is accompanied by an increase in Rad52 foci (Fig. 6b), denoting an accumulation of recombinogenic DSBs, and thus confirming a defect in HDR under replication-stress and DNA damage-inducing conditions. Since *las17-14* and *las17-aid* (+IAA) mutants exhibited similar phenotypes for the DNA-damage read-outs of our interest, we employed *las17-14* mutant for all subsequent studies.

Next, since accumulating evidence indicate that a natural source of replication stress and DNA damage is the formation of ectopic DNA-RNA hybrids (R loops[39,40]), and we have previously shown that WASp deficiency leads to accumulation of such structures[11] and that RPA facilitates ribonuclease H1 (RNH1) action in suppressing ectopic R loops[41], we tested if Las17 has a role in preventing R loop accumulation in yeast. Indeed, the *las17-14* mutant accumulates significantly more R loops at two previously validated genes, as determined by DNA-RNA immunoprecipitation (DRIP) using the antibody S9.6. The specificity of DRIP-signal was further verified by in vitro treatment with RNH1, which eliminated the DRIP-signals (Fig. 6c). Importantly, accumulation of DNA breaks as determined by Rad52 foci were also suppressed by RNH1 over-expression in vivo (Fig. 6d), confirming that R loops in these mutants are a natural source of replication stress and DNA damage. This implies that Las17, like human WASp[11], is required to maintain a healthy R loop balance and prevent R-loop-mediated DSBs in yeast.

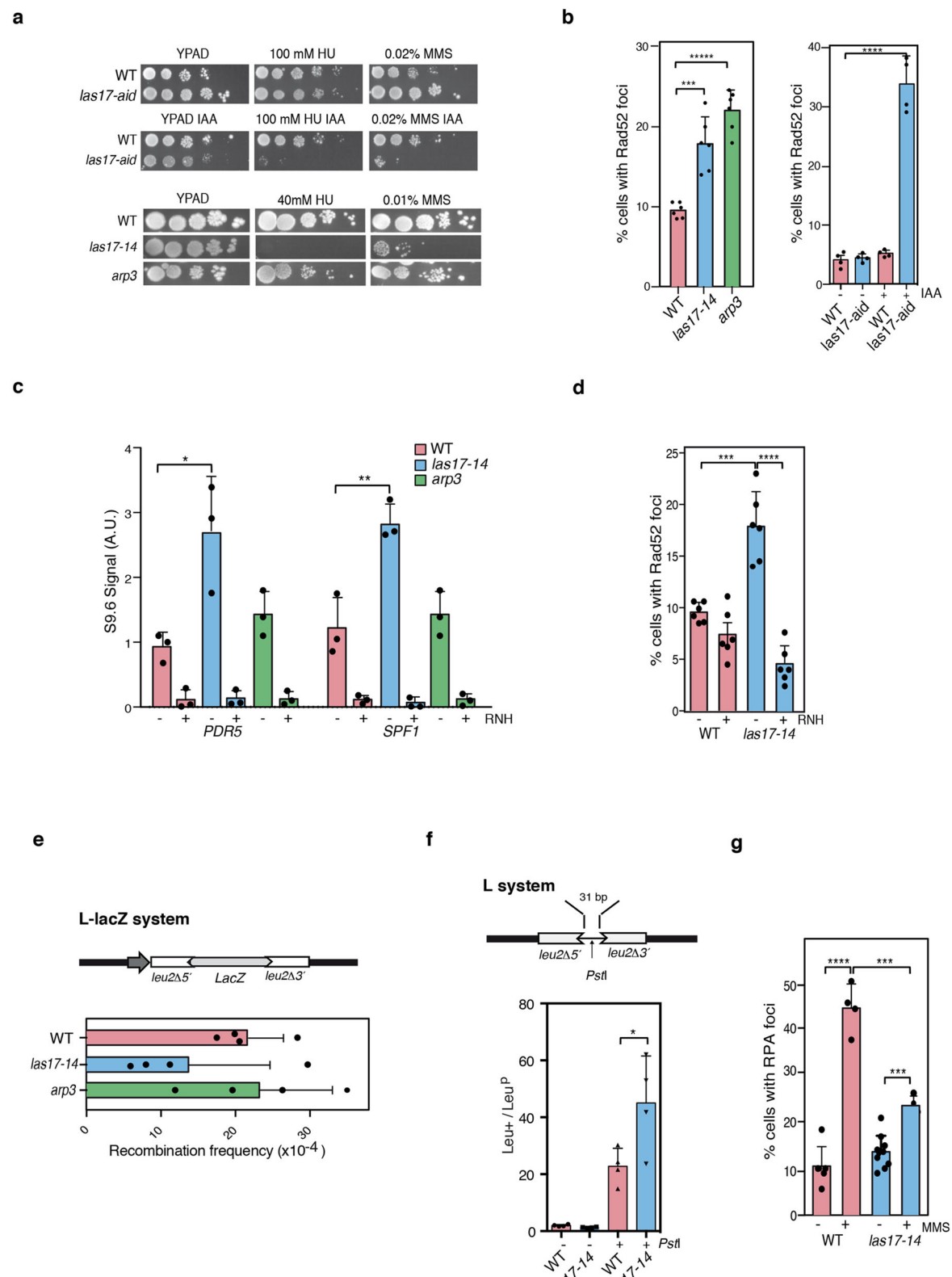

Next, to directly test how the Las17 protein may influence HDR, we analyzed recombination by single-strand annealing deletion events in the plasmid-born recombination system L-lacZ based on two 0.6-kb *leu2* repeats with a 3-kb *lacZ* sequence in between (Fig. 6e). Notably, we found that the frequency of recombination in *las17-14* mutant was not higher than that in wild-type yeast (Fig. 6e), which we would have expected if the

high levels of Rad52 foci observed in these *las17* mutants were a consequence of an increase in DSBs (Fig. 6b). Instead, these results imply that DSBs are not efficiently processed by HDR in the *las17-14* mutant, and consequently Rad52 foci accumulate at high levels. To formally test this possibility, we determined the capacity of the *las17* mutants to repair DSBs by single-strand annealing (SSA), a repair pathway that does not depend on the

**Fig. 6 Las17-deficiency renders *S. cerevisiae* hypersensitive to genotoxins. a**. 10-fold serial dilution assays of the indicated yeast strains with and without 2 mM of auxin (IAA), in plates containing hydroxyurea (HU) and methyl methanesulfonate (MMS) at the indicated dose or concentration. **b**. Percentages of nuclei with Rad52 foci in indicated strains (WT, *las17-14*, *las17-aid*, *arp3*), cultured in media with or without auxin (IAA) addition. Mean +SEM, n = 6 independent experiments. \*\*\**p* < 0.001, \*\*\*\**p* < 0.0001 by Paired Student´s t-test. **c**. Enrichment signals of DNA-RNA immunoprecipitation (DRIP) using S9.6 antibody at the indicated gene loci of the yeast genome, detected by qPCR in WT, *las17-14* and *arp3* mutant cells. For the RNase H1 (RNH) control sample, half of the DNA was treated in vitro with 8 μl of RNase H overnight at 37 °C. Mean +SEM, n = 3 independent experiments. \**p* < 0.05, \*\**p* < 0.01 by Paired Student´s t-test. A.U. denotes arbitrary units. **d**. Percentage of nuclei with Rad52 foci in WT and *las17-14* mutant with (+) or without (-) RNase H1 (RNH) overexpression in vivo. Mean+SEM, n = 5 independent experiments. \*\*\**p* < 0.001, \*\*\*\**p* < 0.0001 by Paired Student´s t-test. **e**. Recombination frequencies of WT and *las17-14* and *arp3* mutants transformed with pRS314 L-LacZ carrying a *leu2* direct-repeat system and cultured in selective medium. Mean+SEM, n = 4 independent experiments. **f**. Ratio of Leu+ recombinants (SSA events):LeuP non-recombinants (papilators on SC-Leu reflecting original plasmids containing both repeats) arising after transformation of WT and *las17-14* strains with uncut and cut (DSB) pRS316-L plasmid carrying the *leu2* direct-repeat system. The cut site created by *Pst*I is located between the repeats. Mean+SEM, n = 4 independent experiments, \**p* < 0.05 by Paired Student´s t-test. Light boxes represent *leu2Δ3'* and *leu2Δ5'* truncated alleles of *LEU2* used as 600-bp direct repeats. **g**. Percentage of nuclei with RPA foci in WT and *las17-14* mutant with (+) or without (-) MMS treatment. Mean+SEM, n = 5 independent experiments. \*\*\**p* < 0.001, \*\*\*\**p* < 0.0001 by Paired Student´s t-test.

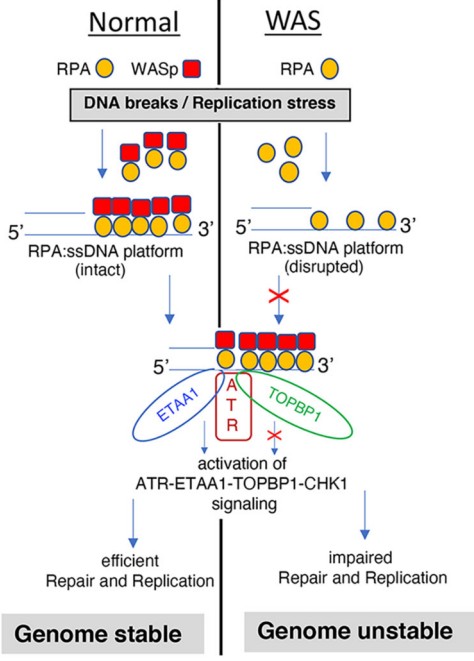

**Fig. 7 Graphical abstract of WASp:RPA alliance in genome stability.** Shown are the sequential steps in the HR pathway involved in DNA repair or replication reactions, and the effect of WASp deficiency on RPA-coupled mis-regulation of these critical steps/reactions, which ultimately lead to genome instability, spontaneously or insult-induced.

Rad51 strand exchange protein for homologous recombination. For this, we generated a DSB between the two 0.6-k *leu2* direct-repeats harbored by a CEN-plasmid at a unique *Pst*I site (Fig. 6f) and determined the level of repair by SSA. We show that *las17*-14 mutant displays a significantly higher level of DSB-induced SSA (cut) as determined by the ratio of Leu+ recombinants versus LeuP non-recombinants (Fig. 6f) compared to the spontaneous levels (uncut), which implies that Las17 deficiency favors repair of the breaks by SSA. It is worth noting that arp3 yeast mutant carrying the mild *arp3-D11A* mutation shows similar levels of both sensitivity to genotoxins and spontaneous recombination as wild-type organism, but higher levels of Rad52 foci (Fig. 6a, b), suggesting increased spontaneous breaks and a more complex role of arp2/3 in HDR, as reported previously[12]. Importantly, and like the RPA defect observed in human immune cells (Figs. 1a, 3), damage-induced RPA foci formation is reduced also in the MMS-treated *las17-14* mutant (Fig. 6g).

Together, human and yeast data suggest a defect in RPA loading/retention at DNA damaged sites as causative of altered DSB-repair (Fig. 7), consistent with an evolutionarily conserved role of WASp in the DDR and RSR.

## Discussion

Our study demonstrates that WASp is a required component of the DNA stress-resolution apparatus, with an obligatory role in the formation/stabilization of the RPA:ssDNA-complex, a crucial step in the DNA repair and replication cascades[33,34]. WASp helps coordinate a core RPA function of binding ssDNA. Because RPA innately binds ssDNA tightly, "chaperons" like WASp are needed to modulate the RPA:ssDNA-complex and thus improve the efficiency of protein-DNA "handoffs" needed to complete DNA repair (Fig. 7). This genome-stabilizing role of human WASp is conserved down to yeast, in which Las17 deficiency also cause RPA dysfunction and DNA damage. Collectively, our findings in human lymphocytes and yeast models propose that WASp:RPA signal integration likely co-evolved to support physiologic DNA replication, RSR, and DSB repair, whose integrity is fundamental to organismal fitness and in the prevention of disease[42]. Since homologous recombination (HR) is an essential mechanism for efficient DNA replication and DSB repair (homology-directed repair, HDR)[43–45], and since we show that Las17-deficient cells favor DSB repair via the SSA pathway, which is relatively mutagenic[45], our findings propose mis-regulated HR/HDR as an important driver of genome instability in WASp-deficient cells in both higher and lower eukaryotes, and a mechanism for onco-genesis in human WAS. Since WASp is expressed in all hematopoietic-derived cell lineages, the deleterious effects on the genome we have uncovered in WASp-deficient lymphocytes could also potentially manifest in other immune cells. We therefore propose that the degree of combinatorial defects in both the adaptive and innate immune cells contribute to the development of different clinical severity phenotypes in different WAS patients.

From a cell-biological standpoint, since nuclear Arp2/3 and F-actin are essential in DNA repair and ATR-linked RSR[12,46], our results propose a region-specific, dual-actions of WASp in genome-stability, one that is RPA-mediated (via a small segment of A-region of the VCA-domain) and another F-actin-mediated (via rest of the VCA-domain). Hence, the VCA-domain function of WASp, historically known only for supporting F-actin cortical-cytoskeletal remodeling, extends to DNA replication and repair in the nucleus. Moreover, due to the intrinsic nature of WASp to oligomerize through VCA:VCA-domain co-association[25], our results suggest a unique molecular mechanism for focally concentrating the enrichment of both Arp2/3:F-actin (via multiple

VC-regions) and RPA (via multiple A-regions) at DNA-damage/ RS sites where WASp accumulates. As such, WASp oligomerization provides a means to enrich multiple RPA-DBDs to ssDNA, the latter resulting in a more stable RPA:ssDNA complexation needed for efficient DNA repair[32]. Conversely, in the steady-state, allosteric autoinhibitory conformation of WASp, which buries the VCA-domain in the body of WASp, provides a mechanism to prevent constitutive/persistent activation of both F-actin- and RPA-linked signaling in the nucleus. These findings suggest a model in which localized actin (oligomeric or polymeric) assembly at RFs provides a pool of RPA at the right time (RSR-induced) and at the right place (stressed RFs, e.g., slow-moving or arrested). Because WASp has a role in modifying chromatin[3,4], our findings do not rule out the possibility that nuclear WASp may influence RSR also by regulating the activity of chromatin-remodelers at stressed RFs. To wit, disruption of actin dynamics at the nuclear envelop[46], by causing global chromatin reorganization, could also indirectly contribute to RF dysfunction in WAS cells. Furthermore, since our yeast studies revealed that *las17* deficiency results in a more pronounced damage phenotype compared to *arp3* deficiency, propose that WASp (Las17) role in genome stability is both Arp2/3-dependent and -independent. It should be noted that since scLas17 has not yet been shown in the nucleus of budding yeast, Las17 may influence these events directly or indirectly, the latter for example by altering the mono/oligomeric vs. polymeric actin balance in the nucleus. Accordingly, WASp is part of the rheostat dial that regulates the measured activation of the RSR and DDR by dynamically modulating RPA-activity at the RF, likely by multiple mechanisms.

From a clinical standpoint, our study proposes WAS as a genotoxin-sensitive, genome-destabilizing immunodeficiency disorder that predisposes to cancer development, in part by mis-regulated RSR and DDR from the RPA defect. Although, loss-of-function of the RPA1-subunit is not causally-linked to cancer development in humans, *RPA1* haplo-insufficient mice develop lymphoid cancers[47], and somatic *RPA1* mutations are found in a subset of cancers[48]. Because genome-instability is a hallmark of cancer, the findings described here support a previously proposed model that WASp functions within the "tumor-suppressor" apparatus in normal lymphocytes[49], and that its loss triggers replication dysfunction and genome-instability contributing to oncogenesis in WAS. Notably, immunodeficiencies frequently occur in certain congenital diseases arising from mutations in essential replication genes[50]. In this connection, multiple *WAS* mutations/ variants (p.D485G; p.E486K; p.D489N; p.D493N; p.D497E; pD497Y) are reported in the public WAS-databases that occur in/or around RBM1, which could potentially disrupt RPA1-activity directly. Moreover, disease-causing mutations are found in the nuclear-localizing and nuclear-export sequences (NLS and NES) located in the N-terminal region of WASp[6], which can potentially affect RPA1-activity by perturbing entry and/or retention of WASp in the nucleus. Thus, establishing the cause-and-effect relationships between patient-derived *WAS* alleles and RPA-linked defects in DNA replication/repair will provide insightful models that can be predictive of patient disease-severity and clinical outcomes.

## Methods

**Cells**. Human CD4 T and CD19 B lymphocytes were employed. T cell line (ND1-WT, HTLV1-immortalized) was established from the PBMCs of a normal donor by MACS cell separation (Miltenyi Biotec, Germany). To generate an isogenic WASp-deficient T cell-line pair (ND1-WKO), we performed CRISPR-mediated KO of *WAS*, as described previously[11]. Briefly, 2 mg of WASP-CRISPR/CAS9-GFP plasmid (sc-400712-KO-2) (Santa Cruz biotechnology, Santa Cruz, CA) was transfected into ND1-WT T cells using Amaxa® Cell Line Nucleofector® Kit V (Lonza Biotec, USA). FACS-sorted GFP+ cells were seeded into 96-well plates by

serial dilution and screened by PCR to identify clones exhibiting complete *WAS*-KO. Successful WASp depletion was verified by RT-PCR for mRNA expression and Western blot for protein expression, and only such *WAS* KO T cell clones were expanded and used for downstream assays. Immortalized Epstein-Barr virus (EBV)-transformed B-lymphoblastoid cell lines (BLCLs), control BLCL harboring WT *WAS* (Cont.3719; ND03719) and WAS patient-derived BLCL harboring E133K (WAS03; ID00003) mutation, were purchased from Coriell Institute (Camden, NJ). These cell lines were periodically tested for mycoplasma contamination.

**Genotoxins**. For human T and B cells, we employed hydroxyurea (HU), low dose (0.2 mM, known to induce replication stress without DSBs) and high dose (4 mM, known to induce replication stress with DSBs), and camptothecin (CPT) (1mM-5 mM concentration range, known to induce single-ended DSBs that are classically caused by endogenous replication fork lesions). For *S. cerevisiae*, we employed both HU and methyl methanesulfonate (MMS) (0.01% or 0.02% concentration), the latter known to induce replication block linked DSBs.

**Recombinant WASp, RPA, and RPA mutants**. Recombinant human WT*WASp and ΔRBM1*WASp mutant proteins were commercially produced employing proprietary methods (GenScript USA Inc., Piscataway, NJ). Briefly, for recombinant human WT*WASp protein, codon-optimized *WAS* gene was cloned into NdeI-HindIII site of pET30a vector (pET30a-WT*WASp). For generating recombinant human ΔRBM1*WASp protein, *WAS* gene from pET30a-WT*WASp plasmid was cut out by NdeI-HindIII digestion and then cloned into NdeI-HindIII site of pET30a vector after RPA1-binding motif (RBM1, amino acids 493-502 of the VCA-domain) was deleted by site-directed mutagenesis (pET30a-ΔRBM1*WASp). Plasmids (pET30a-WASp*WT or pET30a-ΔRBM1*WASp) were transfected into E. coli DE3, and transformed colonies were screened to identify correct colonies by restriction enzyme digestion and sequencing. A single colony was used for large scale culture for protein generation after expression optimization and verification. Target protein was purified using Flag-tag by BacPower™ Customized Protein Service in E. coli expression system (SC1318, GenScript). The protein purity and their molecular weights were determined by standard SDS-PAGE and Western blot. Integrity of target protein was confirmed by LC-MS/MS and peptide mapping. Human recombinant proteins RPA1 (TP302066), RPA2 (TP305715), MAX (TP320343), ARP3 (TP308460), FANCA (TP761743), XRCC2 (TP308330), DNMT1 (TP326414), MYC (TP301611), NF-κB1 (TP308384) and STAT1 (TP313858) were purchased from OriGene (Rockville, MD). ARP2 (ab217837) was purchased from Abcam (Cambridge, MA). Heterotrimeric RPA and RPA1 mutant forms of RPA were expressed as recombinant protein E. coli and them purified over Affi-gel Blue, hydroxylapatite, and Mono-Q columns as described previously[31,51]. Purifications of RPA1-FAB and RPA1-AB were adjusted as described in Ref. [52] (Walther et al. 1999) because of altered elution profiles. The purification of RPA2,3-DwhE substituted DEAE-fastflow for Affi-gel Blue as the first column as described in ref. [31]. All forms of RPA were purified to greater than 95% purity as determined by SDS-PAGE.

**ELISA protein-protein binding assay**. 1 µg of 1st protein diluted in $H_2O$ (volume: 50 µl/well) was coated onto 96 well plate and incubated for 1 h at 25 °C and washed 3 times with washing buffer (1X PBS, 0.2% Tween-20), thereafter blocked with 300 µl of blocking buffer (1X PBS with 5% milk) for 30 min at 25 °C and washed 3 times with washing buffer. 50 µl of 2nd protein diluted in blocking buffer was added, incubated for 1 h at 25 °C, and then washed 3 times with wash buffer. 50 µl of 1:100 dilution of 1st antibody (anti-WASp antibody: proteintech:10987-I-AP, Rosemont, IL) in blocking buffer was added, incubated for 30 min at 25 °C and then washed 3 times with wash buffer. 50 µl of 1:1000 dilution of 2nd antibody (Goat anti-rabbit IgG peroxidase conjugate: Sigma A 6154, St. Louis, MO) in blocking buffer was added, incubated for 30 min at 25 °C and then washed 3 times with washing buffer. 200 µl of freshly prepared 0.8 mg/ml of O-phenylenediamine (Sigma: P5412-50TAB, St. Louis, MO) in 0.05 M phosphate-citrate buffer containing 0.03% sodium perborate (Sigma: P4922-100CAP, St. Louis, MO) was added, incubated at 25 °C for 10 min in dark and then the color change (absorbance) was measured at OD450 nm on microplate reader (FLUOstar omega, BMG LABTECH, Cary, NC). To coat ssDNA-bound 1st protein, 1 µg of 1st protein (RPA) was incubated with 1 nM ssDNA for 20 min at room temperature (final volume 25 µl), and then transferred to 96 well plate for coating.

**Electrophoretic Mobility Shift Assay (EMSA)**. For preparation of nuclear extract (NE), $10 \times 10^6$ cells were harvested, lysed with 400 µl of buffer A (10 mM KCl, 0.2 mM EDTA, 1.5 mM $MgCl_2$, 0.5 mM DTT, and 0.2 mM PMSF) and incubated at 4 °C for 10 min. Lysate was centrifuged for 5 min at $14,000 \times g$. Pellet was resuspended in 100 ml of buffer C (20 mM HEPES [pH 7.9], 420 mM NaCl, 1.5 mM $MgCl_2$, 20% glycerol, 0.2 mM EDTA, 0.5 mM DTT, and 0.2 mM PMSF), incubated at 4 °C for 20 min, centrifuged for 6 min at $14,000 \times g$, and then the supernatants were collected as nuclear extract (NE). Concentration of NE was determined using a precision red advanced protein assay reagent (cytoskeleton ADV02-A, Denver, CO). For preparation of RNA or DNA probes, 20 µl of reaction mixture (2.0 µl RNA, ssDNA, or DNA substrates [2 µM], 2 µl 10x T4-polynucleotide kinase buffer,

1.5 µl T4-polynucleotide kinase [Promega M4108, Madison, WI], 2.5 µl γ-[$^{32}$P]-ATP [BLU502A250UC; 3000 Ci/mmol, 250 µCi/25 µl, Perkin-Elmer, Waltham, MA], 12 µl dH$_2$O) was incubated for 45 min at 37ºC. Unincorporated γ-[$^{32}$P]-ATP was removed by purification on a Nick column (GE Healthcare, Piscataway, NJ) and then total CPM of the radiolabeled probe was measured using liquid scintillation counter (Beckman LS6500, Indianapolis, IN). For EMSA, 25 µl of reaction mixture (5 µg NE, 5 µl 5× gel shift binding buffer [Promega E358A, Madison, WI], dH2O) was incubated at 4ºC for 10 min. Probe (10$^5$ CPM) was added to reaction mixture, incubated at room temperature for 20 min and then resolved on 6% non-denaturing polyacrylamide gels. Gels were dried and subjected to autoradiography. For super-shift assay, 2 µg antibody was added to reaction mixture after the reaction with probe had been completed and incubated at room temperature for 30 min. RPA1 (TA309716) and RPA2 (TA500786) antibodies were purchased from Origene (Rockville, MD), WASp antibody (10987-I-AP) was purchased from Proteintech (Rosemont, IL) and control IgG antibody (sc-66931) was purchased from Santa Cruz Biotechnology (Santa Cruz, CA). All recombinant proteins, antibodies and reagents used in this study are listed in Supplementary Table S1.

**Oligonucleotides, RNA and DNA substrates**. Oligonucleotides used to create ssDNA (61-mer) and dsDNA (61 bp) were adopted from a design reported previously[53]. RNA was chemically synthesized by Integrated DNA Technologies, Inc. (Iowa City, IA) using the same sequence as the 61-mer ssDNA. Sequences of the oligonucleotides used in this study are listed in Supplementary Table S3. Combinations of oligonucleotides annealed to create the DNA substrates used in this study are also listed in Supplementary Table S3. Annealing was performed in a water bath by heating to 94 °C for 2 min, removing from heat and allowing tube to cool to room temperature. The quality of annealing was verified by native gel electrophoresis. Oligonucleotides used to create the 3′-tail (30 bp for the double-stranded part and 31-mer for the single-stranded part), the splayed arm (30 bp for the double-stranded part and 31-mer for the single-stranded part), and the 3′-flap (with a 31-mer flap) were also adopted from a design reported previously[53]. (Supplementary Tables S2 and S3).

**T cell transfection**. For transfection, cells were cultured without antibiotic 1 day before transfection. 3 × 10$^6$ cells were harvested at 90 × g for 10 min and resuspended with 100 µl of Amaxa cell line Nucleofactor Kit V (VCA-1003) or L solution (VCA-1005) (Lonza, Basel, Switzerland). 2 µg of plasmid (WASP-CRISPR/CAS9-GFP for WAS gene knock out, pCMV6-AC-mGFP-WASp-WT or pCMV6-AC-mGFP-WASp-ΔRBM1 for transfection) was added to cell suspension, incubated at room temperature for 10 min, transferred to cuvette and transfection performed using nucleofector 2b device (Lonza, Basel, Switzerland). Transfected cells were cultured for 1 day and then subjected to FACS for sorting GFP-enriched cell population.

**DNA fiber assay**. The assay was performed as previously described[23,24]. Briefly, polyclonal and asynchronous population of human T cells [Wild-type and WASp-deficient (WKO) isogenic pair] or human B cells (normal donor and WAS03 patient-derived) expanded in vitro culture were labelled with 25µM iododeoxyuridine (IdU) (1st label) followed by 250µM chlorodeoxyuridine (CldU) (2nd label) for different durations under genotoxin (HU or CPT)-perturbed conditions or unperturbed control. Finally, 2 µl of labelled cells were placed onto Superfrost glass slides and lysed with 12 µl Lysis buffer (0.2 M Tris–HCl, pH 7.5, 50 mM EDTA, 0.5% SDS) for 12 min at room temperature, and DNA fibers slowly spread onto 15-20° tilted slides, fixed with methanol:acetic acid solution at 3:1 ratio, denatured with freshly prepared 2.5 M HCl for 1 hr at RT, washed with PBS, and then blocked with 37°C pre- warmed 5% BSA in PBS for 30 min. The IdU- and CIdU-labelled DNA was detected by 2-color immunofluorescence after staining with primary (mouse and rat anti-BrDU) and secondary antibodies (Alexa Flour 546 and 488) to detect IdU (red) and CIdU (green), images captured by Zeiss 710 confocal imaging system with 63X magnification, and statistical analyses conducted using Image J software (NIH) or GraphPad software (Prism 8). To convert the fork velocity from µm/min to kb/min, we applied a conversion factor of 2.59 kb/µm. At least 150-200 fiber tracks that were acquired and analyzed per condition. Tracks in large clumps or with overlapping labelling (yellow colored) were excluding from the analyses.

**Western blot and Flow cytometry**. Western blotting assays were performed using commercial reagents shown in Supplementary Table S1. Briefly, protein concentration in the cell lysates was determined by Pierce™ BCA Protein Assay Kit (ThermoFisher), and a set amount was loaded in 4-15% Mini-Protean TGX Precast Gel (Bio-Rad) and transferred to PDVF membrane by Trans-Blot Turbo System (Bio-Rad). The membrane was blocked with TBST-B (10 mM Tris-HCl pH 7.5, 0.5% Tween-20, and 150 mM NaCl, and 5% BSA) for 20 min at RT, and incubated with the primary antibody overnight at 4°C, washed with TBST, and then re-incubated with HRP-conjugated secondary antibody for 1 h at RT. The images of protein bands, developed by incubating the membrane with Clarity Western ECL Substrate (Bio-Rad), were acquired by UVP image system, and gel densitometric analyses performed using Image J software. Cell cycle analyses was performed after

staining 2 × 10$^6$ cells with propidium iodide (PI) at 37 °C for 30 min, and cell-cycle distribution analyzed by flow cytometry (Becton Dickinson LSR II).

**SIRF and PLA**. SIRF assay was performed as described[21]. Briefly, T and B cells, WT and WASp-deficient, grown in log-phase were pulsed with 125 µM EdU at 37°C for 30 min, washed with PBS and then treated with different genotoxins for 4 h. Treated cells were fixed with 4% PFA on a poly-L-lysine pretreated slide, permeabilized with 0.3% Triton X-100, washed with PBS, and then followed by Click-iT reaction. For performing Click-iT reaction, the click cocktail containing 100 mM sodium ascorbate (Sigma Aldrich), 2 mM copper sulfate, and 25 mM biotin-azide in PBS was freshly made for each reaction. The slide was placed in a humid chamber and the click cocktail was added to the slide area containing fixed cells (~30 µl/sample) and incubated at room temperature for 40 min. After the click rection, the slide was washed for 5 min with PBS containing 3% BSA. Primary antibodies were diluted in the permeabilized buffer, applied to individual cell samples, and incubated at 4°C overnight in the humid chamber. Subsequent steps were according to the proximity ligation assay (PLA) per the Duolink In Situ labeling kit PLA protocol from the manufacturer (Sigma Aldrich). Briefly, the slides were washed twice with 80 ml Buffer A (10 mM Tris-HCL pH 7.5, 150 mM NaCl, and 0.05% Tween-20) in a Coplin jar. The slides were then incubated with secondary PLA probes (~25 mL per sample of anti-mouse minus and anti-rabbit plus) in the humid chamber at 37°C for 1 hour. The excess secondary PLA antibodies were removed by tapping the slide, and slides were washed with Buffer A. Ligation mix was freshly prepared by diluting 5X Ligation Stock and 40X ligase in high purity buffer, and applied onto the slides, which were incubated in a humid chamber at 37°C for 30 min. After washing, amplification mix containing Amplification buffer (dilute 1:5) and Polymerase (dilute 1:80) in high quality water was then applied onto the slides and incubated in a humid chamber at 37°C for another 100 min. Finally, Slides were washed with 80 ml Buffer B (0.2 M Tris and 0.1 M NaCl) in a Coplin jar twice, 10 min each, and then with 100X diluted Buffer B for another 1 min. Slides were air dried and applied V-shield mount medium with DAPI and covered with coverslips. Slides were imaged using Zeiss 710 confocal imaging system with 63X magnification, and ~35-40 z-stack images acquired at 0.3 µm step-size were analyzed using combination of Duolink, Fiji Image J, and Prism 8 software.

*iPOND assay*. iPOND assay was performed as described previously[22]. Briefly, T cells (~80 million cells per sample) were incubated with 20 µM EdU in 30 ml culture medium at 37°C for 20 min and then washed with fresh medium at 37°C. For HU or thymidine pluse, cells were resuspended either with 30 ml fresh medium at 37 °C as the control or 4 mM HU or 20 µM thymidine in fresh medium. These cells were further cultured at 37°C for another 2 hrs. After labeling/treatments, each cell samples were washed with PBS, and crossed-linked in 1% Formaldehyde in PBS for 20 min at RT, quenched with glycine at the final concentration of 0.125 M for another 5 min, and washed 2 times in PBS. Cell pellets were permeabilized with 10 ml permeabilizing buffer (0.3% Triton-X/0.5% BSA in PBS) 30 min at RT and washed with 0.5% BSA/PBS. Each cell pellet was resuspended in 10 ml PBS as the control, or 10 ml fresh prepared Click buffer (10 mM Sodium ascorbate, 2 mM CuSO4, and 20 µM Biotin-dPEG7-azide) and incubated for 1-2 hr at room temperature (RT). Cells were washed with 0.5% BSA/PBS and then pellets either frozen at −80°C or immediately used for lysis. Each cell pellet was resuspended in 0.8 ml lysis buffer (25 mM NaCl, 2 mM EDTA, 50 mM Tris pH 8, 1% IGEPAL CA630, 0.2 % SDS, 0.5% sodium deoxycholate, and 1X Halt protease and phosphatase inhibitor cocktail (Thermo Fisher)) and incubated for 10 min on ice. Samples were sonicated with a Branson 250 using the settings, 20-25 W, 20 sec constant pulse, and 40 sec pauses for a total of 4 min on ice. Cell lysates were centrifuged at 18k X g for 10 min at RT. The supernatants were collected and diluted with the dilution buffer (the lysis buffer without SDS or sodium deoxycholate). Streptavidin-agarose beads (Millipore Sigma) (80µl/sample) were washed with the dilution buffer 3 times, and then incubated with the diluted samples overnight at 4°C. The beads were again washed 3 times with RIPA buffer. Captured proteins were separated from beads by incubating beads in 50 µl 2× Laemmli Sample Buffer (Bio-Rad) at 95°C for 25 min. The supernatant was collected, and proteins resolved on 4-15% SDS-PAGE and detected by immunoblotting.

**Neutral COMET assays**. The neutral COMET assays were conducted according to the manufacturer's (Trevigen) specifications, and as we described previously[11]. Briefly, CometSlide (Trevigen, Gaithersburg, Md) plated with cells (2.5 ×10$^5$/mL) were placed in the Sub-Cell GT Agarose Gel Electrophoresis System (Bio-Rad Laboratories, Herçules, Calif) for 45 min at 30 V in neutral electrophoresis buffer. Slides were stained with SYBR-Gold dye, and ~35-50 z-stack images acquired by the Zeiss LSM-710 confocal microscope. Comet images were analyzed for tail-moment statistics calculated using OpenComet software.

**Yeast strains, plasmids, and spotting studies**. Yeast strains used in this study are indicated in Supplementary Table S4. The las17-14 ts allele used is a lack of function mutation that reduces Sec4 stability and membrane binding as the null mutation, whereas arp3-D11A is a ts mutation with mild effects on nucleotide binding[54,55]. The YMK-L17-D strains containing the las17-aid degron allele was

constructed using a yeast strain in which the only copy of the *TIR1* gene was under the constitutive *ADH1* promoter[56] and the plasmid pHyg-AID-9myc[57]. Mid-log cultures were grown in YPAD medium. 10-fold serial dilutions of the culture were prepared with sterile water and 3 µl of each dilution was spotted in solid plates. Plates were incubated 2-3 days at 30 °C.

**Rad52 and RPA foci detection**. Yeast strains were transformed with pWJ1344 and pRS315-GAL:RNH1 or pRS315 plasmids[58–60]. Resulting transformants were grown in glucose- or galactose-containing selective media until exponential growth and then cells were fixed with 2.5% formaldehyde in 0.1 M KHPO pH 6.4 during 10 min and washed twice in 0.1 M KHPO pH 6.6. Afterwards, cells were washed in 0.1 M KHPO pH 7.4 and permeabilized with 80% ethanol during 10 min, followed by resuspension in 1 µg/mL DAPI for staining nuclei. More than 200 nuclei for each experiment were visualized and counted in a fluorescence microscopy Leica DC 350 F microscope. For RPA foci detection, mid-log cultures were grown in YPAD medium with or without 0.005% MMS during 1 hour. More than 200 nuclei for each experiment were visualized and counted in a fluorescence microscopy Leica DC 350 F microscope.

**Recombination and SSA assays**. Single strand annealing (SSA) was determined by the frequency of deletions obtained with the *leu2* direct-repeat recombination systems L or L-lacZ based on *leu2Δ3'* and *leu2Δ5'* truncations of *LEU2* and located in a CEN-URA3 plasmid as described previously[61]. The pRS316-L-lacZ plasmid was introduced into the cells by transformation after which cells were plated on selective SC-ura to isolate independent transformants. Recombination frequencies were calculated as the median value of six independent colonies for each transformation. The average of the median values of four independent transformants was plotted. Recombinants were obtained by plating appropriate dilutions in applicable selective medium. To calculate the total number of cells, these were plated in the same media used for transformation. All plates were grown for 3-7 days at 30 °C.

For DSB-induced SSA, the L system was used. Leu+ SSA deletions or Leu+ papillating colonies (LeuP) carrying the parental direct-repeat construct were determined among the Ura+ transformants by subsequent replica plating onto SC-leu. Experiments were performed under two conditions, either using the intact uncut plasmid or cut with a double-strand break (DSB) in between the repeats before transformation, to determine spontaneous or DSB-induced SSA levels. The efficiency of DSB-induced SSA is shown as the ratio Leu+ SSA recombinants/LeuP non-recombinants

**DNA-RNA immunoprecipitation (DRIP) assay**. DRIP experiments were performed as described[62]. Briefly, cells coming from exponential growth were pelleted by centrifugation and resuspended in 2.4 mL of spheroplasting buffer (1 M sorbitol, 2 mM Tris–HCl pH 8.0,100 mM EDTA pH 8.0, 0.1% v/v β-mercaptoethanol, 2 mg/mL Zymoliase 20 T). Samples were incubated at 30 °C during 30 min. After centrifugation, the pellet was resuspended in 1.125 mL of buffer G2 (0.8 mM Guanidine HCl, 30 mM Tris–HCl pH 8.0, 30 mMEDTA pH 8.0, 5% Tween 20, 0.5% Triton X-100) together with 40 µL of 10 mg/mL RNase A and incubated at 37 °C during 30 min. Then, 75 µL of 20 mg/mL proteinase K were added and samples stood at 50 °C for 1 hour. DNA was purified by chloroform:isoamyl alcohol (24:1) and precipitated with 1 volume of isopropanol. With the help of a glass pasteur pipette, DNA was transferred to a new eppendorf where resuspended in 150 µl of 1X TE (1 mM Tris–HCl pH 7.5, 0.5 mM EDTA pH 8.0) and digested overnight with 50 U of HindIII, EcoRI, BsrGI, XbaI and SspI. Half of the DNA was treated with 8 µl of RNase H (New England BioLabs) overnight 37 °C as RNaseH control. Both samples were incubated with S9.6 antibody- Dynabeads Protein A (Invitrogen) complexes (previously incubated overnight at 4 °C) during 2.5 hours at 4 °C. Samples were then washed 3 times with 1X binding buffer (10 mM NaPO4 pH 7.0, 0.14 M NaCl, 0.05% Triton X-100). DNA was eluted in 100 µl elution buffer (50 mM Tris pH 8.0, 10 mM EDTA, 0.5 % SDS) treated 45 min with 7 µl of 20 mg/mL proteinase K at 55 °C and purified with Macherey-Nagel DNA purification kit. Real-time quantitative PCR was performed using iTaq universal SYBR Green (Biorad) with a 7500 Real-Time PCR machine (Applied Biosystems).

**Reporting summary**. Further information on research design is available in the Nature Research Reporting Summary linked to this article.

## Data availability

The data supporting the findings of this study are available from the corresponding authors upon reasonable request. Source data for the figures and supplementary figures are provided as a Source Data file. Source data are provided with this paper.

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

## Acknowledgements

This work was supported in part by the NIH, National Institute of Allergy and Infectious Diseases (NIAID) grants R01AI146380 (to Y.M.V.), grant A24881 from ICR Intramural Grant and Cancer Research UK Programme (to W.N.), the European Research Council grant ERC2014 AdG669898 TARLOOP and the Spanish Ministry of Science and Innovation grant PDI2019-104270GB-I00/ BMC) (to A.A.), the University of Iowa Dance Marathon (UIDM) research award (to S.S.H), Research Bridge Award from the Carver College of Medicine University of Iowa (to Y.M.V), and the Endowments from the Mary Joy & Jerre Stead Foundation and from the PennState THON & Children's Miracle Network (to Y.M.V). A subset of data was obtained at the Flow Cytometry Facility, which is a Carver College of Medicine/Holden Comprehensive Cancer Center core research facility at the University of Iowa. We thank the UIDM for supporting the research laboratory space where bulk of this work was carried out.

## Author contributions

Y.M.V. and W.N. conceived the project. S.S.H. generated all CRISPR/Cas9-mediated *WAS* gene KO cellular models and WASp domain-deleted mutant, performed ELISA assays, EMSA assays, and cell-cycle flow studies. K.K.W. performed SIRF, PLA, iPOND, Comet assay, Western blots, and DNA fiber assays. M. L. G.-R. performed *Saccharomyces* studies. M. W. provided recombinant RPA proteins and RPA domain-deleted mutants. Y.M.V. and A.A. designed experiments and interpreted the data. Y.M.V. wrote the manuscript with inputs from A.A., and editing contributions from W.N. and M.W. All authors reviewed the manuscript.

## Competing interests

The authors declare no competing interests.
