## [Peer Review File · Nature Communications]

Title: WASp modulates RPA function on single-stranded DNA in response to replication stress and DNA damageREVIEWER COMMENTS

Reviewer #1 (Remarks to the Author):

Han et al.'s manuscript present evidence for the WAS protein function in RPA binding to ssDNA in response to DNA damage. The authors demonstrate that WASp colocalizes with RPA only after DNA damage; purified WASp binds RPA in vitro and the interaction is mediated by specific motif present also in other proteins binding RPA; WASp enhances RPA binding to ssDNA in vitro and in vivo (in response to DNA damage); WASp enhances ATR activation in response to DNA damage (RPA is critical for ATR activation); WASp is important for normal fork progression in response to DNA damage; WASp deficient cells show decreased or delayed DSB repair. Finally, yeast ortholog Las17 seems to have a similar function to WASp. Altogether, RPA is a central protein complex in DNA repair, replication, and recombination, so regulation of RPA is of broad interest. This manuscript presents interesting data suggesting a direct role of WASp in the regulation of RPA binding to ssDNA. The authors should provide more evidence for the "critical" function of WASp in RPA regulation.

Main criticism:

The authors claim that WASp plays a "critical" role in RPA binding to ssDNA. RPA mutants that have decreased ssDNA binding or mutants that have decreased nuclear RPA (*rtt105*) show specific and very characteristic phenotypes in yeast (e.g. Kolodner, Symington and Chen/Zhang labs). An example and straightforward test are to look for large deletions/duplications. To support their central claim on the role of WASp in RPA binding to ssDNA, the authors should test the mutation rates and mutation types using any yeast reporter assay in *las17* mutant. The presence of deletions/duplications would strongly argue in favor of the author's hypothesis. As RPA inhibits annealing of ssDNA, an increase of Alt-EJ or SSA are expected in WASp or *Las17* deficient cells - this could be tested as an alternative way to prove the critical role of these proteins in RPA binding to ssDNA.

Yet another way to show WASp specificity in promoting RPA regulation would be overexpression of RPA. In yeast work presented here, *Rnh1* o/e is shown to rescue the *las17* mutant. Overexpression of RPA should rescue WASp or *Las17* deficiency and would provide more direct evidence for the function of these proteins, specifically in RPA regulation. O/e of RPA was done by many labs and should not be a problem. Notably, o/e of RPA rescues many phenotypes of *rtt105* mutant.

Proteins characterized so far in yeast and frogs suggest that *Rtt105* and XRIP promote nuclear localization of RPA, respectively. Does WASp impact on RPA nuclear localization?

Minor

Rtt105 in yeast play an important role in RPA binding to ssDNA. Therefore it would be interesting to establish the consequences of eliminating both *Rtt105* and *Las17* in yeast. The human ortholog of *Rtt105*, called hRIP- α , was proposed to play a similar role in RPA regulation (PMID: 34140406) so testing it along with WASp depletion could reveal possible redundant functions.

Fig. 4c. It would be essential to add a control Δ lataRBM1*WASp to show that RPA-WASp interaction is needed for activation of the ATR mediated checkpoint.

Does Las17 carry an RPA-interaction domain similar to WASp?

Page 5 "WASp directly binds RPA in vitro ad in vivo"

As all the work here is done with purified protein, it is not clear why the authors claim binding "in vivo".

Page 8 - It's hard to follow the experiments done with DBD-F/A/B and other RPA mutants without going to published work. These mutants should be introduced better for a general audience.

The recent manuscript analyzed in detail phenotypes in *rtt105* mutant related to deficiency of RPA. It should be cited (PMID: 34140406).

Is the overall cellular level of RPA after DNA damage the same in WT and WASp deficient cells?

Reviewer #2 (Remarks to the Author):

In the manuscript by Han et al., the authors use PLA, SIRF and EMSA assays to demonstrate a role for WASp in enhancing RPA-ssDNA formation and activation of ATR signaling in response to replication stress. The authors identify an RPA-binding motif in WASp, and they show that WASp-deficient lymphocytes fail to establish RPA-ssDNA at stalled forks and have unstable forks using DNA fiber assays. Finally, the authors demonstrate conservation of this function of WASp in yeast mutant strains. The manuscript is well-done and the results support the overall conclusions. Furthermore, the findings are interesting and of importance to the field. There are a few minor concerns that should be addressed before acceptance of the manuscript for publication.

Minor points

1) The authors rely heavily on PLA and SIRF methods. While these are very useful assays, they do have some caveats and are not always as reliable or quantitative as immunofluorescence (IF) imaging. Can the authors validate some of their findings using IF? For example, data in Figure 3a showing a decrease in RPA S33 at replication forks can be shown by conventional IF methods. Similarly, it would be helpful to use IF to visualize WASp at stalled replication forks. This would allow the authors to see if WASp accumulates in foci in a timecourse study.

2) How quickly does WASp accumulate at stalled replication forks? Does it localize early in the replication stress response or is it a late response? Given WASp role to stabilize RPA and promote ATR

signaling, I'd suspect WASp has early roles; however, the authors use HU and CPT for 4 hours throughout the manuscript, which is well after stress activates ATR. Shorter timecourse studies are preferred when identifying a new replication stress response factor.

3) In figure 4c, is the effect on CHK1 phosphorylation reproducible in the B-cell model of WASp deficiency?

4) In Figure 5a-d, is the effect on fork stability reproducible in the B-cell model of WASp deficiency?

5) In Figure 5e the data show that WKO have increased comet tail moments in unstressed cells, and that stress doesn't further increase this. In the text, the authors ignore this point and instead highlight the increased damage "under genotoxic stress". The data seem to argue against this and implicate DNA breaks arising from something other than replication stress and failure to activate ATR. The micronucleus formation data in Figure 5f similarly suggest a more complex defect in the WKO cells, particularly in B cells versus T cells. Given the data in Figure 1 suggesting WASp doesn't associate with γ H2AX or RPA in unstressed cells, it is unclear how the comet assay and parts of the micronucleus assay fit. Is it possible that other WASp functions (e.g. preventing R-loops or promoting HDR) could underlie this data? Please discuss this further.

6) The authors refer to an "accompanying paper" by Nieminuszczy et al throughout the manuscript as evidence that this function of WASp occurs in non-immune cells. At this point, this seem inappropriate or at the least premature unless this other paper has already been accepted for publication.

Reviewer #3 (Remarks to the Author):

The paper by Seong-Su Han et al addresses the question why mutations in mammalian WASp lead to genome instability. They argue that Wasp binds RPA, and stabilizes its interaction with ssDNA, at least in mammalian cell exposed to certain DNA insults. Their conclusions, and extension of these to yeast Las17, argue along several disparate lines that do not hold together in a coherent logical argument. Indeed, I cannot decipher the logical flow of the data nor of the experimental line, and the conclusions/model proposed in the discussion (i.e. that the defect arising from WASp misregulation reflects altered G-/F-actin ratios) is inconsistent with what they appear to show here – that WASp binds stalled forks, stabilizes RPA, promotes R-loops (presumably the non-annealed strand binds the RPA) and then ultimately controls checkpoint kinase response - basically becoming a nuclear co-factor of repair. In contrast to the "repair factor" scenario, the G-/F-actin ratio mechanism agrees with earlier work from this laboratory which argued that WASp leads to nucleosome remodeler defects (many repair remodelers have G-actin as a subunit, thus this is consistent with the G/F actin shift and with the fact that short actin polymers inhibit remodelers in vitro). While I am not sure how that reflects the WASp-RPA interaction they seem to document, it does seem to jive better with known cellular functions of WASp, which is a plasma membrane actin focus-forming protein that regulates actin filament bifurcation through Arp2/3. The model the propose does not require at all that WASp is associated with damage

(which they spend 2 figures on – yet it is an interaction I am not convinced is real). Indeed, while the G-/F-actin imbalance conclusion seems more reasonable to this reviewer, it seems to contradict much of the current data in this paper shows. Thus, there are such major discrepancies in the manuscript that I cannot actually grasp their logic.

In conclusion, while the question is legitimate and interesting, this set of data cannot be published without extensive additional studies, and a rethinking of the argument they want to make. As for the yeast work, it should be omitted entirely, as it presents a mishmash of distinct DNA damage conditions (DSBs, stalled forks, bypass synthesis due to MMS and R-loop resolution), which represent nonoverlapping pathways of repair. Indeed, Rad52 foci are not involved in most of these repair pathways, yet they argue that Rad52 foci is a main las17-14 phenotype. The yeast work is poorly controlled, and besides the random damage conditions used, las17-14 is an inappropriate Las17 mutant (a degron Las17 should be used). Note that they obtain the same effects with loss of Arp2/Arp3 complex, which Las17/WASp regulates to generate actin filament forking. It must be clearly stated that loss of Las17 – ARP interaction (and disruption of Arp2/Arp3) generate strong primary phenotypes due to near complete lack of endocytosis, exocytosis, mitochondrial integrity, loss of cytoskeletal integrity as well as remodeler malfunction and F/G actin imbalance! In any case, the “hand-wave” at yeast adds nothing of value to the data presented.

Here are specific details and controls this reviewer finds missing:

1. In the abstract the authors suggest that the human and yeast data are parallel if not identical, yet I do not think they are at all. The same assays are not done and as pointed out below, the range of damage introduced into yeast has little to do with the conditions studied in human cells.
2. Why should stabilization of RPA of ssDNA cause increased degradation of Replication forks? and even if it did, why would that increase R-loops which require RNA synthesis and a failure to process and export it? The links here are tenuous and illogical to me.
3. On CPT they see more G1 phase arrest – not replication fork arrest – thus this must represent yet another pathway of repair. RPA foci are not generally correlate with CPT induced damage as there is no extensive resection. The CPT phenotype appears disconnected with their RPA hypothesis.
4. The PLA and SIRF assays depend on rolling circle amplification and are as such extremely sensitive to primary and secondary antibody purity and lack of cross reaction. In fact the rolling circle amplification will show strong signals even with almost no protein present. Thus the data in Figure 1 and 2 need additional controls. If the authors wish to claim colocalization of WASp with stalled forks, then they need to tag WASP and show WITHOUT rolling circle amplification that the protein is actually found at forks. CHIP with appropriate controls (using the same antibody on untagged strains) is necessary to be able to claim this
5. What does the role of WASp at stalled forks have to do with Double strand break resection, which was earlier published as a WASp dependent event (Jean Gautier, and Irene Chiolo labs)? These links are unclear, tenuous and suggest another more general pathway, such as remodeler malfunction.
6. The yeast data is a chaotic mixture of different insults, assays and mutants, and the logic of what it means is unclear. It is not the case that yeast Las17 moves to the yeast nucleus under DNA damaging conditions, rather it forms aggregates that colocalize with mitochondria. That being said, the HU sensitivity of las17-14 might well reflect a total collapse of all Fe-S protein synthesis, as many repair and replication factors have Fe-S centers. Fe-S centers are totally dependent on mitochondrial integrity and

HU inhibits their synthesis. Controls for Fe-S protein integrity are needed. In addition, HU does not form Double strand breaks (it is used as a fork stalling agent that specifically does not form DSBs), set somehow HU sensitivity is lumped together with Rad52 foci (recombination foci). Finally, the R-loop phenotype is a complete outlier in this argument. Why should loss of Las17 (or inactivation of its interaction with Arp2/3) create R-loops, in the context of their model ? The arguments range from Rad52 foci (homology dependent recombination) to mismatch repair and fork stalling. In any case, the model simply argues, loss of WASp/Las17 leads to genome instability (by an unknown mix of problems) and since that was already known, there is need for this paper.

7. The model suggests that WASp is necessary for checkpoint kinase (ATR) activation and there seems to be less ATR at damage foci in WASp deficient cells but ATR is still activated. It is simple to monitor in yeast as well – and I am quite sure they will find Mec1 activated. The impact of Las17/WASp on checkpoint responses need to be quantitatively analysed in all circumstances.

8. Finally, the biggest problem I have with this study and the localization of WASp is that WASp/Las17 is a highly disordered protein and has been one of the classable “phase separating” proteins studied (e.g. by Mike Rosen). That means that it is highly insoluble, and responds to minor changes in salt, hydrophobicity and other internal conditions, forming aggregates. It may well shift from its normal localization in foci at the plasma membrane to the nucleus during the overnight incubation, but that does not mean it does this in a physiologically relevant manner. Thus, the detection of some level of WASp at ssDNA is not too surprising, but it does not speak to its abundance nor does it argue per se for a repair mechanism involving WASp. The paper desperately needs the addition of assays checking what happens to the cytoplasmic pool of WASp under damage conditions, tracking the protein with an epitope tag under normal and damaged conditions, without the PLA or SIRF amplification event. Without a physiological involvement at the break, it is likely that the WASp phenotypes are triggered in trans. Such controls are necessary – both in mammalian and in yeast cells.

9. A direct test of the actin imbalance hypothesis is needed – that is, the paper needs the promised but missing G vs F actin quantitation along with a clearing out of the other hypotheses.

March 22, 2022

We thank all reviewers for providing excellent and insightful comments/suggestions to further strengthen the key message of our manuscript.

We have **addressed all key criticism** with multiple additional experiments.

Point-by-point responses:

Reviewer- 1

Han et al.'s manuscript present evidence for the WAS protein function in RPA binding to ssDNA in response to DNA damage. The authors demonstrate that WASp colocalizes with RPA only after DNA damage; purified WASp binds RPA in vitro and the interaction is mediated by specific motif present also in other proteins binding RPA; WASp enhances RPA binding to ssDNA in vitro and in vivo (in response to DNA damage); WASp enhances ATR activation in response to DNA damage (RPA is critical for ATR activation); WASp is important for normal fork progression in response to DNA damage; WASp deficient cells show decreased or delayed DSB repair. Finally, yeast ortholog Las17 seems to have a similar function to WASp. Altogether, RPA is a central protein complex in DNA repair, replication, and recombination, so regulation of RPA is of broad interest. This manuscript presents interesting data suggesting a direct role of WASp in the regulation of RPA binding to ssDNA. The authors should provide more evidence for the "critical" function of WASp in RPA regulation.

Main criticism:

The authors claim that WASp plays a "critical" role in RPA binding to ssDNA. RPA mutants that have decreased ssDNA binding or mutants that have decreased nuclear RPA (rtt105) show specific and very characteristic phenotypes in yeast (e.g. Kolodner, Symington and Chen/Zhang labs).

- 1. An example and straightforward test are to look for large deletions/duplications. To support their central claim on the role of WASp in RPA binding to ssDNA, the authors should test the mutation rates and mutation types using any yeast reporter assay in las17 mutant. The presence of deletions/duplications would strongly argue in favor of the author's hypothesis. As RPA inhibits annealing of ssDNA, an increase of Alt-EJ or SSA are expected in WASp or Las17 deficient cells - this could be tested as an alternative way to prove the critical role of these proteins in RPA binding to ssDNA.*

Response: Following the reviewer's suggestion we performed experiments in which we generated a DSB in between two direct repeats and analyzed the frequencies of SSA events in Wt. and Las17 mutant. We show that there is a significant increase in the frequency of DSB-induced SSA events (new Fig. 6f). Thus, our studies strongly support our key conclusions, i.e., WASp is critical for RPA:ssDNA binding activity in the RSR and/or DNA repair.

- 2. Yet another way to show WASp specificity in promoting RPA regulation would be overexpression of RPA. In yeast work presented here, Rnh1 o/e is shown to rescue the las17 mutant. Overexpression of RPA should rescue WASp or Las17 deficiency and would provide more direct evidence for the function of these proteins, specifically in RPA regulation. O/e of RPA was done by many labs and should not be a problem. Notably, o/e of RPA rescues many phenotypes of rtt105 mutant.*

Response: We did try to address this; however, we faced several challenges. First, we overexpressed the three subunits of RPA from a GAL1 promoter by generating strains in which the three RPA constructs were stably integrated in chromosomes (a gift from Belén Gómez-González and John Diffley), but such an overexpression caused synthetic lethality. To circumvent this challenge, we repeated the experiments using low amount of glucose to reduce the levels of overexpression to see whether we can manage some conditions in which the cells are still able to grow. Unfortunately, cells still did not grow. We conclude that overexpression of RPA under these conditions cause lethality. It is worth noting that in a previously reported study, in which overexpression of RPA in yeast cells from plasmids and from a weaker promoter (Ruff et al 2016, Cell Reports 17:3359-68), there was no certainty about the levels of overexpression in those conditions (as reported by the authors). Furthermore, they argue that plasmid loss could have favored growth of the subpopulation not overexpressing RPA.

3. *Proteins characterized so far in yeast and frogs suggest that Rtt105 and XRIP promote nuclear localization of RPA, respectively. Does WASp impact on RPA nuclear localization?*

Response: To address this question, we chose to perform confocal IF imaging, since this study would allow us to visualize on a single cell basis if RPA1 was ectopically “trapped” in the cytoplasmic compartment in WASp-deficient (WKO) T cells. We find that the RPA1 nuclear signal in WKO T cells is visually comparable to WT T cells, in the steady state in vitro culture conditions. This suggests that human WASp is not required for RPA1 nuclear localization (Supplementary Fig S2h)

4. *Minor: Rtt105 in yeast play an important role in RPA binding to ssDNA. Therefore, it would be interesting to establish the consequences of eliminating both Rtt105 and Las17 in yeast. The human ortholog of Rtt105, called hRIP- α , was proposed to play a similar role in RPA regulation (PMID: 34140406) so testing it along with WASp depletion could reveal possible redundant functions.*

Response: This is an interesting biological question, especially for the yeast system; but is beyond the scope of this work, and perhaps tangential to testing/or establishing a novel role for human WASp in RPA-linked genome stability. Follow up experiments will first have to test whether hRIP-a is expressed in human T and B lymphocytes or not, and if so, whether tools exist to deplete it without detrimental effects on the lymphocytes.

5. *Fig. 4c. It would be essential to add a control delta-RBM1*WASp to show that RPA-WASp interaction is needed for activation of the ATR mediated checkpoint.*

Response: Agree. We did this. Our data show that selective deletion of the RPA-binding motif (RBM1) of WASp is sufficient to cause decreased Chk1 (pSer345) activation, which is suggestive of an impaired ATR-mediated checkpoint activation (new Fig. 4e).

6. *Page 5 "WASp directly binds RPA in vitro ad in vivo" As all the work here is done with purified protein, it is not clear why the authors claim binding "in vivo".*

Response: the reviewer is correct, in that PLA can only indicate very close proximity (< 10 nm distance) of 2 proteins, which we interpret as “association”. We have now reworded this sentence in the text.

7. *Does Las17 carry an RPA-interaction domain similar to WASp?*

Response: Yes, it does. We show the consensus alignment of yeast motif with that of human WASp in Fig. 2b under WASp orthologs.

8. *Page 8 - It's hard to follow the experiments done with DBD-F/A/B and other RPA mutants without going to published work. These mutants should be introduced better for a general audience.*

Response: We provide appropriate references (Ref # 31, 51), in which their generation functional characterization is detailed.

9. *The recent manuscript analyzed in detail phenotypes in rtt105 mutant related to deficiency of RPA. It should be cited (PMID: 34140406).*

Response: Indeed, this is an impressive paper, which we had cited in the first version. We do so again in the revised version (now Ref #17 by Wang et al PNAS 2021).

10. *Is the overall cellular level of RPA after DNA damage the same in WT and WASp deficient cells?*

Response: The reviewer's question gets to the need to rule out any quantitative differences in the total expression of RPA in WASp-deficient cells. We did this assay and show comparable level of RPA1 in both WT and WASp-deficient T cells, at steady-state and after replication stress (new supplementary Fig. S2g).

Reviewer- 2

Reviewer #2 (Remarks to the Author):

In the manuscript by Han et al., the authors use PLA, SIRF and EMSA assays to demonstrate a role for WASp in enhancing RPA-ssDNA formation and activation of ATR signaling in response to replication stress. The authors identify an RPA-binding motif in WASp, and they show that WASp-deficient lymphocytes fail to establish RPA-ssDNA at stalled forks and have unstable forks using DNA fiber assays. Finally, the authors demonstrate conservation of this function of WASp in yeast mutant strains. The manuscript is well-done and the results support the overall

conclusions. Furthermore, the findings are interesting and of importance to the field. There are a few minor concerns that should be addressed before acceptance of the manuscript for publication.

Minor points

11. The authors rely heavily on PLA and SIRF methods. While these are very useful assays, they do have some caveats and are not always as reliable or quantitative as immunofluorescence (IF) imaging. Can the authors validate some of their findings using IF? For example, data in Figure 3a showing a decrease in RPA S33 at replication forks can be shown by conventional IF methods. Similarly, it would be helpful to use IF to visualize WASp at stalled replication forks. This would allow the authors to see if WASp accumulates in foci in a timecourse study.

Response: The reviewer rightly cautions us about the limitations of PLA and SIRF, which are both IF imaging approaches that visualize WASp association with forks at the single cell level. To improve rigor, we now show WASp association with the forks using iPOND, a time-tested assay done at a cell-population level (Figs. 1c, 3b). Moreover, iPOND/Western blot allows gel densitometric quantitation of protein enrichment at the forks upon stress relative to unstressed control. We now provide this data and their p values.

12. How quickly does WASp accumulate at stalled replication forks? Does it localize early in the replication stress response or is it a late response? Given WASp role to stabilize RPA and promote ATR signaling, I'd suspect WASp has early roles; however, the authors use HU and CPT for 4 hours throughout the manuscript, which is well after stress activates ATR. Shorter time course studies are preferred when identifying a new replication stress response factor.

Response: Excellent suggestion. We conducted iPOND assay at 30min (early) and 4hr (late) after HU stress to directly test whether WASp is a *bona fide* RSR factor. The reviewer was correct in predicting that WASp might have an early role in RSR (new Fig. 1c).

13. "In figure 4c, is the effect on **CHK1 phosphorylation reproducible in the B-cell model of WASp deficiency?**"

Response: In direct response to this question, we conducted the experiment and find that the CHK1 defect seen in WASp-deficient T cells is reproducible also in WASp-deficient B cells (new Fig. 4d).

14. "In Figure 5a-d, is the **effect on fork stability reproducible in the B-cell model of WASp deficiency?**"

Response: Similarly, we conducted the suggested experiment and find that the fork defects seen in WASp-deficient T cells is reproducible also in WASp-deficient B cells (new supplementary Fig. S4).

15. In Figure 5e the data show that WKO have increased comet tail moments in unstressed cells, and that stress doesn't further increase this. In the text, the authors

ignore this point and instead highlight the increased damage “under genotoxic stress”. The data seem to argue against this and implicate DNA breaks arising from something other than replication stress and failure to activate ATR. The micronucleus formation data in Figure 5f similarly suggest a more complex defect in the WKO cells, particularly in B cells versus T cells. Given the data in Figure 1 suggesting WASp doesn’t associate with γ H2AX or RPA in unstressed cells, it is unclear how the comet assay and parts of the micronucleus assay fit. Is it possible that other WASp functions (e.g. preventing R-loops or promoting HDR) could underlie this data? Please discuss this further.

Response: The reviewer’s interpretation of our comet tail findings in chemically unstressed WKO cells is correct. We have previously shown that WASp-deficient T cells accumulate “bad” R loops, and hence also show high frequency of R-loop-mediated DSBs. (ref # 11). Gautier’s group showed WASp role in the early step of HDR (ref # 12). Moreover, the in vitro culture condition itself imposes a certain level of stress, which may be handled better in the presence of WASp. Accordingly, WASp-deficient cells will manifest spontaneous genome-instability, reported in a higher frequency of comet tail moment relative to normal, even at the steady-state. Clastogens and replication stressors inflict further “insult-on-injury” creating a highly unstable genome. We include this explanation in the text.

16. *The authors refer to an “accompanying paper” by Nieminuszczy et al throughout the manuscript as evidence that this function of WASp occurs in non-immune cells. At this point, this seem inappropriate or at the least premature unless this other paper has already been accepted for publication.*

Response: we have omitted this reference.

Reviewer- 3

17. *The paper by Seong-Su Han et al addresses the question why mutations in mammalian WASp lead to genome instability. They argue that Wasp binds RPA, and stabilizes its interaction with ssDNA, at least in mammalian cell exposed to certain DNA insults. Their conclusions, and extension of these to yeast Las17, argue along several disparate lines that do not hold together in a coherent logical argument. Indeed, I cannot decipher the logical flow of the data nor of the experimental line, and the conclusions/ model proposed in the discussion (i.e. that the defect arising from WASp misregulation reflects altered G-/F-actin ratios) is inconsistent with what they appear to show here – that WASp binds stalled forks, stabilizes RPA, promotes R-loops (presumably the non-annealed strand binds the RPA) and then ultimately controls checkpoint kinase response - basically becoming a nuclear co-factor of repair.*

In contrast to the “repair factor” scenario, the G-/F-actin ratio mechanism agrees with earlier work from this laboratory which argued that WASp leads to nucleosome remodeler defects (many repair remodelers have G-actin as a subunit, thus this is consistent with the G/F actin shift and with the fact that short actin polymers

inhibit remodelers in vitro). While I am not sure how that reflects the WASp-RPA interaction they seem to document, it does seem to jive better with known cellular functions of WASp, which is a plasma membrane actin focus-forming protein that regulates actin filament bifurcation through Arp2/3.

The model the propose does not require at all that WASp is associated with damage (which they spend 2 figures on – yet it is an interaction I am not convinced is real).

Indeed, while the G-/F-actin imbalance conclusion seems more reasonable to this reviewer, it seems to contradict much of the current data in this paper shows. Thus, there are such major discrepancies in the manuscript that I cannot actually grasp their logic.

In conclusion, while the question is legitimate and interesting, this set of data cannot be published without extensive additional studies, and a rethinking of the argument they want to make.

Response: To address this point, we employed additional rigor in testing the interaction of WASp with forks. Accordingly, we now show both by SIRF (at single cell level) and iPOND (at cell-population level) that WASp inducibly accumulates at stressed forks (Figs. 1b, 1c). In our PLA studies, because WASp co-accumulates in-vivo/in-situ with both γ H2A.X (a marker of DNA damage and replication stress) and phosphorylated RPA2 (Ser33) (a marker of ATR-mediated DNA damage response activation), we conclude that WASp dynamically associates with the DNA damage (or replication stress) sites in both T and B cells (Fig. 1a).

We should also point out that our experiments were not designed to formally address how the relative abundance of G-actin vs. F-actin levels in the nucleus might impact a nucleosome-mediated defect in DNA repair, nor do we make such a claim in this manuscript. However, since the reviewer brought up this connection, and since my lab has previously established an essential role of WASp in the activity of SWI/SNF (BAF) complexes at the gene promoter loci (Ref # 4), we now include this possibility in our discussion section.

18. As for the yeast work, it should be omitted entirely, as it presents a mishmash of distinct DNA damage conditions (DSBs, stalled forks, bypass synthesis due to MMS and R-loop resolution), which represent nonoverlapping pathways of repair.

Response: there are 2 compelling reasons to include yeast work:

1. What the yeast data is showing is that the novel biological paradigm uncovered in the human cellular models is conserved down to yeast. Providing such a validation across species is demonstrative of a strong scientific rigor and renders the novel biological model more generalizable.
2. Linking WASp role to managing DNA damage or replicative stress whether mediated by exogenous genotoxins or endogenous transcriptional R loops, suggest a more fundamental role of WASp in genome stability.

19. *“Indeed, Rad52 foci are not involved in most of these repair pathways, yet they argue that Rad52 foci is a main Las17-14 phenotype”.*

Response: Rad52 loading and Rad52 foci formation typically monitor recombinogenic DSBs in yeast. As such, yeast Rad52 homolog has an essential role in the SSA repair pathway (but not in Alt-EJ), and as indicated to reviewer #1 we now show that the frequency of DSB-induced SSA is higher in Las17-deficient strain, which fits well with our proposed model. This result clearly supports the view that normal homologous recombination requiring strand invasion is less efficient, so that when a DSB is generated in between two repeats, SSA repair pathway is more efficient. Therefore, our Rad52 foci results indicate that in the Las17 mutants (verified also in Las17-AID degenon) more DNA breaks occur that are repaired via the SSA repair pathway rather than standard recombination.

20. *The yeast work is poorly controlled, and besides the random damage conditions used, Las17-14 is an inappropriate Las17 mutant (a degenon Las17 should be used).*

Response: Reviewer's suggestion to verify our key findings in a Las17-aid degenon strain is appropriate; it certainly increases the experimental rigor of the yeast studies. We have generated a degenon strain of Las17 confirmed by Western (new Suppl Fig S6). This strain also shows the expected phenotypes, i.e., sensitivity to HU and MMS (Fig. 6a), as well as an increase in DNA breaks detected by Rad52 foci (new Fig. 6b). Since the results in Las17-aid degenon were the same as Las17-14 mutant, the findings validate the use of the ts Las17-14 mutant for the rest of the study.

21. *Note that they obtain the same effects with loss of Arp2/Arp3 complex, which Las17/WASp regulates to generate actin filament forking. It must be clearly stated that loss of Las17 – ARP interaction (and disruption of Arp2/Arp3) generate strong primary phenotypes due to. near complete lack of endocytosis, exocytosis, mitochondrial integrity, loss of cytoskeletal integrity as well as remodeler malfunction and F/G actin imbalance. In any case, the “hand-wave” at yeast adds nothing of value to the data presented.*

Response: In Fig. 6 (yeast studies), we show a clear difference in the damage phenotypes between Arp3 deficiency and Las17 deficiency alone, as it relates to growth (Fig. 6a), R loop accumulation (Fig. 6c), and recombination frequency (Fig. 6e). Although, either deficiency alone causes damage-induced growth impairment, Las17 deficiency shows a more pronounced damage phenotype compared to arp3 deficiency. Moreover, R-loop accumulation is also more pronounced in Las17 deficiency relative to arp3. This distinction is important to make, since these findings suggest that WASp (Las17) role in genome stability is both Arp2/3 -dependent and -independent. Accordingly, the yeast studies are essential for making this biologically important distinction. We include this discussion in the text.

22. *"In the abstract the authors suggest that the human and yeast data are parallel if not identical, yet I do not think they are at all."*

Response: In the abstract, we state that the deficiency of both WASp in human cells and *scLas17* in yeast manifest in RPA recruitment defect to the forks and genome instability. This conclusion is well supported by the provided experimental evidence. The reviewer rightly cautions us that similar outcome in 2 disparate systems/organisms should not be considered as "identical", which we agree, and hence have purposefully avoided the use of the term (identical) in the abstract.

23. *Why should stabilization of RPA of ssDNA cause increased degradation of Replication forks? and even if it did, why would that increase R-loops which require RNA synthesis and a failure to process and export it ? The links here are tenuous and illogical to me.*

Response: Sorry, we are unable to understand this question. WASp-deficiency results in decreased or unstable RPA:ssDNA complexation, which renders the RFs vulnerable to genotoxin-induced damage. The reviewer somehow interprets our SIRF, iPOND, PLA, and EMSA results to indicate increased "stabilization" rather than "destabilization" of RPA:ssDNA complexes.

24. *On CPT they see more G1 phase arrest – not replication fork arrest – thus this must represent yet another pathway of repair. RPA foci are not generally correlate with CPT induced damage as there is no extensive resection. The CPT phenotype appears disconnected with their RPA hypothesis.*

Response: The reviewer is perhaps alluding to the relationship between the type of damage, cell cycle stage, and repair pathway choices. If so, the reviewer is correct in that end resection is suppressed in G1 and active in S/G2. By the same token, it is also true that damaging agents (e.g., UV, HU, CPT) that stall or break RFs will generate high levels of RPA-ssDNA complexes resulting in RPA activation (RPA2 phosphorylation at Ser33) (Refs #33, 34). In this context, we show that in WASp-deficient cells, RPA2 (pSer33) foci formation following DNA damage is impaired. (Fig. 3a). These results provide an additional support for a role of WASp in RPA-mediated genome stability, and its loss (in WAS patients) results in genome instability linked to impaired RPA function, whether due to HU or CPT.

25. *The PLA and SIRF assays depend on rolling circle amplification and are as such extremely sensitive to primary and secondary antibody purity and lack of cross reaction. In fact the rolling circle amplification will show strong signals even with almost no protein present. Thus the data in Figure 1 and 2 need additional controls. If the authors wish to claim colocalization of WASp with stalled forks, then they need to tag WASP and show WITHOUT rolling circle amplification that the protein is actually found at forks. CHIP with appropriate controls (using the same antibody on untagged strains) is necessary to be able to claim this.*

Response: The reviewer rightly cautions us about the limitations of the imaging-based assays (PLA and SIRF). In response, we now provide additional verification using iPOND to report on WASp and RPA enrichment frequencies at stressed forks (new Figs. 1c, 3d). Furthermore, we have included PLA and SIRF imaging data for T and B cells without any damage, and for the iPOND assays, we have included WKO T cells, which show no WASp signals. These serve as

our internal experimental controls that rules out any spontaneous background signals and also point to the specificity of antibodies used (Figs. 1a, 1b, 3b).

26. “What does the role of WASp at stalled forks have to do with Double strand break resection, which was earlier published as a WASp dependent event (Jean Gautier, and Irene Chiolo labs)? These links are unclear, tenuous and suggest another more general pathway, such as remodeler malfunction.

Response: Gautier’s work showed WASp role in the early “pre-repair” step of the DSB repair, i.e., physical transport of the DSB ends to the HDR repair zones. Our present work has greatly expanded this role, by demonstrating WASp role in the “actual repair reactions” in the HDR pathway by modulating RPA activity. Disrupting RPA activity at the stalled forks can lead to genome instability including via DNA strand breaks.

27. The yeast data is a chaotic mixture of different insults, assays and mutants, and the logic of what it means is unclear. It is not the case that yeast Las17 moves to the yeast nucleus under DNA damaging conditions, rather it forms aggregates that colocalize with mitochondria. That being said, the HU sensitivity of las17-14 might well reflect a total collapse of all Fe-S protein synthesis, as many repair and replication factors have Fe-S centers. Fe-S centers are totally dependent on mitochondrial integrity and HU inhibits their synthesis. Controls for Fe-S protein integrity are needed. In addition, HU does not form Double strand breaks (it is used as a fork stalling agent that specifically does not form DSBs), yet somehow HU sensitivity is lumped together with Rad52 foci (recombination foci).

Response: The Fe-S is an interesting point, but as far as we know from the published literature, there is no unequivocal evidence for this relationship. We have worked in the past with mutants of the Fe-S clusters, and we found that the phenotypes are varied, not leading necessarily to HU sensitivity or increase in damage as high as the ones we report here. On the other hand, Fe-S protein integrity is not a measure of the direct impact on replication. The Fe-S clusters also affect other DNA repair steps, such as those of NER, which in yeast involves Rad3 that also has Fe-S. However, the phenotype of Rad3 does not affect replication nor does it produce HU sensitivity. Thus, it is not possible to generalize that Fe-S protein integrity leads to the phenotypes described in this article. Hence, the *Las17-14* and degron phenotypes reported here cannot be explained by an Fe-S protein collapse. We would add in any case that any difference in Fe-S protein integrity that could be observed in any mutation causing HU sensitivity, even in bona-fide replication factors or DNA damage checkpoint factors if analyzed at some point (something that to our knowledge has not been done extensively), would be more likely the result of a pleiotropic effect on cell cycle progression, ROS accumulation and other subcellular components that would indirectly affect Fe-S cluster. This would be an interesting project for the future, but it is beyond the scope of this study.

28. Finally, the R-loop phenotype is a complete outlier in this argument. Why should loss of Las17 (or inactivation of its interaction with Arp2/3) create R-loops, in the context of their model?”

Response: We previously published that WASp deficiency, by impairing the activity of topoisomerase 1 (TOP1), provokes accumulation of pathological (bad) R loops in human T cells (ref 11). In this study, we show that yeast lacking Las17 also accumulate R loops. This way we consolidate WASp role in R loop maintenance, and one that is evolutionarily conserved.

Bad R loops are known to cause genomic instability through multiple mechanisms, such as through their cleavage by TC-NER factors and transcription-replication conflicts. Accordingly, our model proposes WASp as an essential component of the apparatus that maintains genome stability, be it through maintaining a health R loop balance, mitigating replication stress, and/or being part of DNA repair process via the HDR pathway (as proposed by Gautier's group).

In addition, as better explained now in the text, different studies have shown that R loops are a common source of replication stress and DNA breaks. So, we show that in *Las17* mutants this is also the case, since the lack of Las17 causes the accumulation of R loops as well as the high levels of Rad52 foci, which are suppressed by RNH1 overexpression. Therefore, this result confirms that Las17 is required to solve spontaneous DNA damage, including that generated by R loops. Indeed, this is consistent with previous work in human cells showing that the Fanconi Anemia repair pathway is also required to deal with DNA breaks generated by R loops, as a consequence of which FA-defective cells accumulate R loops (Scwhab et al, Mol Cell 2015; García-Rubio et al, PLoS Genet 2015).

29. *The arguments range from Rad52 foci (homology dependent recombination) to mismatch repair and fork stalling. In any case, the model simply argues, loss of WASp/Las17 leads to genome instability (by an unknown mix of problems) and since that was already known, there is need for this paper.*

Response: We provide multiple experimental evidence that uncovers an essential role of human WASp and yeast *scLas17* in RPA functions at the fork, loss of which results in replicative stress, fork collapse, and genomic instability. This finding is new for the field, since it describes the requirement of WASp-dependent RPA role in the **actual repair reactions** at the DNA damage/stressed fork sites. Gautier's work, on the other hand, showed the requirement of WASp-dependent Arp2/3 role in transporting DNA break ends to repair zones. This is a **pre-repair step**. Accordingly, together, our 2 papers provide a more comprehensive view of WASp roles in genome stability.

30. *The model suggests that WASp is necessary for checkpoint kinase (ATR) activation and there seems to be less ATR at damage foci in WASp deficient cells but ATR is still activated. It is simple to monitor in yeast as well – and I am quite sure they will find Mec1 activated. The impact of Las17/WASp on checkpoint responses need to be quantitatively analysed in all circumstances.*

Response: Agree. As noted by the reviewer, our data show that WASp loss does not completely shut off ATR activation, rather it diminishes this check point response. In multiple western blot assays (Figs. 4c, d, e), we show quantitative differences in the magnitude of pChk1 expression relative to total Chk1 in WASp-deficient cells, and in cells expressing WASp mutant lacking the motif that binds with RPA1. We conclude that WASp is an important (but not the sole) factor regulating ATR check point activation under DNA damaging/ RS conditions. Indeed, ATR is recruited to stress forks by its binding partner ATRIP, with the latter requiring RPA-ssDNA for binding, As such, the lower level of RPA at stressed forks in WASp-deficient cells would negatively impact on the level/stability of ATRIP:ATR complexes at stressed forks.

31. Finally, the biggest problem I have with this study and the localization of WASp is that WASp/Las17 is a highly disordered protein and has been one of the classable “phase separating” proteins studied (e.g. by Mike Rosen). That means that it is highly insoluble, and responds to minor changes in salt, hydrophobicity and other internal conditions, forming aggregates. It may well shift from its normal localization in foci at the plasma membrane to the nucleus during the overnight incubation, but that does not mean it does this in a physiologically relevant manner. Thus, the detection of some level of WASp at ssDNA is not too surprising, but it does not speak to its abundance nor does it argue per se for a repair mechanism involving WASp. The paper desperately needs the addition of assays checking what happens to the cytoplasmic pool of WASp under damage conditions, tracking the protein with an epitope tag under normal and damaged conditions, without the PLA or SIRF amplification event. Without a physiological involvement at the break, it is likely that the WASp phenotypes are triggered in trans. Such controls are necessary – both in mammalian and in yeast cells.

Response: Mike Rosen is a giant in the field of WASp/N-WASp biology, with whom I have collaborated since he and I were at Sloan-Kettering. The reviewer is likely referencing one of his works published in eLife, 2014. Here, Rosen group concludes that “interactions between multivalent proteins could be a general mechanism for cytoplasmic adaptor proteins to organize membrane receptors into micrometer-scale signaling zones”. While Rosen’s group has employed in vitro and lipid bilayer artificial systems, our studies are done in vivo, esp. in the intact cells. In fact, our first discovery of WASp nuclear localization and its key chromatin-resident function was made in primary human cells freshly isolated from normal donor peripheral blood (the most physiologic of the systems). Subsequently numerous other laboratories have duplicated our findings in multiple other systems/organisms including the work of the Nobel Laureate John Gurdon from UK showing Wave protein location and function in the nucleus of *Xenopus* oocytes. As such, Rosen’s group also mention a similar mechanism at play during in vitro interactions between protein and nucleic acid. Thus, our findings align with Rosen’s overall conclusion that WASp creates a micrometer scale signaling zones, wherever it is located (at the plasma membrane, in Golgi, or nucleus). Note, WASp has a functional NLS and NES, and thus is a cytoplasmic-to-nucleus shuttling protein. Importantly, there are a number of disease-causing mutations in the NLS and NES, which is demonstrative of a clinically significant function of WASp in the nucleus of the bone-marrow derived cells where it is most abundantly expressed.

32. A direct test of the actin imbalance hypothesis is needed – that is, the paper needs the promised but missing G vs F actin quantitation along with a clearing out of the other hypotheses.

Response: This is not the hypothesis that we have proposed anywhere in the current manuscript. Nonetheless, this is an interesting model to test in the future, given the role of WASp in nucleating F-actin.

33. The paper desperately needs the addition of assays checking what happens to the cytoplasmic pool of WASp under damage conditions, tracking the protein with an epitope tag under normal and damaged conditions, without the PLA or SIRF

amplification event. Without a physiological involvement at the break, it is likely that the WASp phenotypes are triggered in trans. Such controls are necessary – both in mammalian and in yeast cells.

Response: Good point. We conducted western blots to address this question (supplementary Fig. S3). We show that the amount of WASp present in the nucleus roughly doubles (from ~20% enrichment to ~50%) following HU-induced stress. Because WASp contains functional nuclear localizing (NLS) and nuclear exit (NES) motifs, we cannot say if the observed nuclear WASp accumulation is solely due to increased cytoplasmic-to-nuclear transport (i.e., increased nuclear entry) or increased nuclear retention (i.e., reduced nuclear exit) or both. Either way, this new data further consolidates WASp's nuclear role in replication stress/DNA damage signaling.

REVIEWERS' COMMENTS

Reviewer #1 (Remarks to the Author):

The revised manuscript is improved. Increased SSA is a new genetic evidence for the role of Las17 in regulation of RPA loading. However, the weakness of the SSA assay used here is that it was done using cut plasmid transformation, and so there are additional variables and it is hard to predict at what point ssDNA forms and annealing occurs. Unfortunately, RPA overexpression experiments were not successful. The authors note in responses:

“It is worth noting that in a previously reported study, in which overexpression of RPA in yeast cells from plasmids and from a weaker promoter (Ruff et al 2016, Cell Reports 17:3359-68), there was no certainty about the levels of overexpression in those conditions (as reported by the authors)”

This is not correct as the manuscript states clearly that at least Rfa1 was overexpressed 3.6x: “Because the antibodies used to detect RPA only recognize the large subunit, we cannot state that all three subunits are overexpressed, only that Rfa1 is expressed at 3.6-fold higher level than in the empty vector strain.”

Despite these weaknesses and considering all body of work taken together the manuscript should be considered for publication. It presents interesting observations.

Reviewer #2 (Remarks to the Author):

In the revised manuscript by Han et al., the authors performed additional experiments to address each of my concerns raised in the original submission. These include the addition of iPOND to solidify WASp association with replication forks, the incorporation of shorter exposure to HU illustrating an early role of WASp, and the confirmation of key experiments in the B-cell model. In sum, the conclusions presented in this manuscript are well-supported by the experiments and data collected, and I recommend it for publication.

Reviewer #3 (Remarks to the Author):

The revision of the paper "WASp modulates RPA function on single-stranded DNA ..." by Han et al has been improved in several aspects. The authors have reinforced their error-prone amplification data that mapped WASp to ssDNA at stalled forks with iPOND data, which reinforces the colocalization proposed in the previous version. In the revised paper, the conclusions are less overinterpreted with respect to mammalian WASp, and I am convinced that mammalian WASp interacts with RPA (at least in vitro) and likely stabilizes ssDNA at R-loops in B- or T-cells. It should be made more clear that WASp is almost only

expressed in lymphocyte lineage, and that this phenomenon may be B-/T-cell specific. That fact should probably be included in the title (i.e. that this is likely to be lymphocyte specific; see attached data on analysis of WASp expression among mammalian tissues). It would have been helpful to state the expression pattern of WASp in the introduction.

Major problems still exist with respect to the interpretation of the LAS17 data (LAS17 is a *S. cerevisiae* protein with partial homology to WASp).

I do not doubt the results they present (that *las17* mutation or ablation sensitizes cells to replication stress and other types of DNA damage and may favor SSA over HR). The fact that LAS17 is a key regulator of DNA damage stress response has been amply demonstrated in earlier publications (the authors are encouraged to read Rochette et al., *J. of proteomics* 100, p 25 - 36, 2014). In that study it is shown that *las17-14* is hypersensitive to MMS, and the mutation strongly affects a large number of protein-protein interactions throughout the cell. It has been amply demonstrated in a number of papers that LAS17 coordinates the cellular response to both mechanical and DNA damage stress, although its mode of action is not clear (indeed it affects a large number of Protein-protein interactions). However, the desired interpretation of Han et al, i.e., that budding yeast Las17 acts like the mammalian WASp in lymphocytes, that is, by binding RPA at stressed replication forks, is not a compelling hypothesis for the following reasons.

1. S.c. LAS17, unlike WASp, does not have a NLS. The region that separates the N terminal from the C-terminal protein binding domains in WASp is not conserved in Las17, and the region that contains the NLS in mammalian WASp, is not present. Moreover, the exportin signal and exportin itself do not appear to be conserved.
2. Various high throughput studies have failed to see any relocation of Las17 to the nucleus in yeast cells exposed to either HU or MMS (see Tkach et al., *Nature Cell Biology* 14, 966pp , doi:10.1038/ncb2549). In this study a thorough analysis of proteins that change location upon DNA damage (HU and MMS) is carried out and LAS17 does not relocate to the nucleus. There is also no evidence from any other labeling or GFP fusion studies that LAS17 is found in the nucleus of budding yeast.
3. LAS17-GFP does not show any nuclear localization, and extensive studies from my own laboratory (which are unfortunately not published) have used hypersensitive DAM-ID methods to detect LAS17 in the nucleus under various conditions of stress, but failed to detect any increase in response to damage. This, combined with the fact that LAS17 does not have a nuclear import nor export signal, makes it very unlikely that it is stabilizing an abundant factor like RPA at ss DNA regions in the nucleus.

It is clear that the authors cannot accept an argument based on unpublished data. However, they must at the very least, entertain the argument that the loss of *las17* (the *ts* mutation *las17-14* or Las17 degradation) acts in an indirect manner, affecting Mec1 activation and checkpoint response by altering a large number of protein-protein interactions.

Again in unpublished data, it was shown that LAS17 forms aggregates near the Golgi or ER on Zeocin,

which induces single- and double-strand breaks, and abasic sites. Loss of las17 strongly affects survival of all types of stress, not only those that induce DNA damage. Thus the authors must at the very least entertain the fact that what they convincingly demonstrate for mammalian WASp is not conserved in yeast.

My hypothesis is that the replication fork / DNA damage repair defects in yeast stemming from the loss of LAS17 reflect alterations in G/F actin ratios. There is no doubt that loss of Las17 changes this, as it is the key activator of Arp2/3 and stimulates actin filament formation on its own (as an actin chaperone). However, in budding yeast there is no evidence for actin filaments in the nucleus (unlike those found in mammalian cells under stress). Instead, in yeast the function of actin dependent remodelers are compromised upon alteration of G/F actin balance, and this likely contributes to the repair phenotypes observed.

The authors overinterpret their yeast data in manner aligned with their lymphocyte observations. I do not doubt what they measure in T- and B- cells, nor do I doubt the strong effect of las17 depletion or mutation on repair in yeast. But if LAS17 protein were acting by stabilizing RPA then its deficiency should be at least partially compensated by RPA overexpression. This experiment did not yield the desired results.

In my opinion, without direct evidence (e.g. CHIP data for LAS17 at R-loops in yeast), their interpretation of Rad52 foci, R-loop increase and reagent sensitivity, can be explained by a variety of mechanisms that do not require the presence of Las17 in the cell nucleus. I think the yeast data should be omitted, but at the very least, their preferred interpretation must be softened by considering the fact that LAS17 has never been shown to be nuclear in budding yeast, and it lacks the necessary NLS for nuclear relocation.

May 20, 2022

Point-by-point responses:

Reviewer- 1

The revised manuscript is improved. Increased SSA is a new genetic evidence for the role of Las17 in regulation of RPA loading. However, the weakness of the SSA assay used here is that it was done using cut plasmid transformation, and so there are additional variables and it is hard to predict at what point ssDNA forms and annealing occurs. Unfortunately, RPA overexpression experiments were not successful. The authors note in responses:

“It is worth noting that in a previously reported study, in which overexpression of RPA in yeast cells from plasmids and from a weaker promoter (Ruff et al 2016, Cell Reports 17:3359-68), there was no certainty about the levels of overexpression in those conditions (as reported by the authors)”

This is not correct as the manuscript states clearly that at least Rfa1 was overexpressed 3.6x: “Because the antibodies used to detect RPA only recognize the large subunit, we cannot state that all three subunits are overexpressed, only that Rfa1 is expressed at 3.6-fold higher level than in the empty vector strain.”

Despite these weaknesses and considering all body of work taken together the manuscript should be considered for publication. It presents interesting observations.

Response: Thank you for this clarification. Accordingly, we provide the following correction to our sentence: “As such, in a previously reported study, in which overexpression of RPA (Rfa) in yeast cells from plasmids and from a weaker promoter (Ruff et al 2016, Cell Reports 17:3359-68), the level of expression at least of Rfa1 subunit was reported to be several fold higher than the control vector, however, due to reagent/antibody limitations the authors were unable to definitively conclude if all three subunits of RPA were similarly overexpressed or not”.

Reviewer- 2

In the revised manuscript by Han et al., the authors performed additional experiments to address each of my concerns raised in the original submission. These include the addition of iPOND to solidify WASp association with replication forks, the incorporation of shorter exposure to HU illustrating an early role of WASp, and the confirmation of key experiments in the B-cell model. In sum, the conclusions presented in this manuscript are well-supported by the experiments and data collected, and I recommend it for publication.

Response: Thank you.

Reviewer- 3

1. The revision of the paper "WASp modulates RPA function on single-stranded DNA ..." by Han et al has been improved in several aspects. The authors have reinforced their error-prone amplification data that mapped WASp to ssDNA at stalled forks with iPOND data, which reinforces the colocalization proposed in the previous version. In the revised paper, the conclusions are less overinterpreted with respect to mammalian WASp, and I am convinced that mammalian WASp interacts with RPA (at least in vitro) and likely stabilizes ssDNA at R-loops in B- or T-cells. It should be made more clear that WASp is almost only expressed in lymphocyte lineage, and that this phenomenon may be B-/T-cell specific. That fact should probably be included in the title (i.e. that this is likely to be lymphocyte specific; see attached data on analysis of WASp expression among mammalian tissues). It would have been helpful to state the expression pattern of WASp in the introduction.

Response: Thank you for bringing up the issue related to cell lineage-specific WASp expression. We agree that this is important to clarify.

Both in the abstract and introduction, we do state that the effect of WASp deficiency on RPA activity/function is in human lymphocytes (*Abstract*). As such, in the first paragraph of *Introduction*, we state that the effect of WASp on genome stability is observed in human T and B lymphocytes. Moreover, in the first paragraph of *Discussion*, we link our key findings to human lymphocytes.

It should be noted, however, that WASp is expressed in **all** cell types of the hematopoietic lineage, i.e., WASp expression is not restricted to the lymphocytes only. WASp is expressed in NK cells, monocytes, macrophages, neutrophils, megakaryocytes, dendritic cells, etc. Accordingly, we now add the following statement in the *Discussion*:

"Because WASp is expressed in all hematopoietic-derived cell lineages, the deleterious effects on the genome we have uncovered in WASp-deficient lymphocytes could also potentially manifest in other immune cells. We therefore propose that the degree of combinatorial defects in both the adaptive and innate immune cells contribute to the development of different clinical severity phenotypes in different WAS patients."

We should point out that WASp was recently shown to be expressed also in non-hematopoietic cell lineages, i.e., murine fibroblasts and in human bone cancer cells (osteosarcoma). This published work (PMID: 29925947), challenged the long-held view that WASp expression is limited only to the bone-marrow derived cell lineages. Of note, these cell lineages have high expression of neural-WASp (N-WASp), a member of the WASp-family of proteins. Since N-WASp is more ubiquitously expressed relative to WASp, and because N-WASp also has a nuclear location and function, future research is required to clarify unique vs. redundant roles of WASp and N-WASp in replication stress responses and DNA repair.

2. Major problems still exist with respect to the interpretation of the LAS17 data (LAS17 is a *S. cerevisiae* protein with partial homology to WASp).

I do not doubt the results they present (that las17 mutation or ablation sensitizes cells to

replication stress and other types of DNA damage and may favor SSA over HR). The fact that LAS17 is a key regulator of DNA damage stress response has been amply demonstrated in earlier publications (the authors are encouraged to read Rochette et al., J. of proteomics 100, p 25 - 36, 2014). In that study it is shown that las17-14 is hypersensitive to MMS, and the mutation strongly affects a large number of protein-protein interactions throughout the cell. It has been amply demonstrated in a number of papers that LAS17 coordinates the cellular response to both mechanical and DNA damage stress, although its mode of action is not clear (indeed it affects a large number of Protein-protein interactions).

However, the desired interpretation of Han et al, i.e., that budding yeast Las17 acts like the mammalian WASp in lymphocytes, that is, by binding RPA at stressed replication forks, is not a compelling hypothesis for the following reasons.

1. S.c. LAS17, unlike WASp, does not have a NLS. The region that separates the N terminal from the C-terminal protein binding domains in WASp is not conserved in Las17, and the region that contains the NLS in mammalian WASp, is not present. Moreover, the exportin signal and exportin itself do not appear to be conserved.

2. Various high throughput studies have failed to see any relocation of Las17 to the nucleus in yeast cells exposed to either HU or MMS (see Tkach et al., Nature Cell Biology 14, 966pp , doi:10.1038/ncb2549). In this study a thorough analysis of proteins that change location upon DNA damage (HU and MMS) is carried out and LAS17 does not relocate to the nucleus. There is also no evidence from any other labeling or GFP fusion studies that LAS17 is found in the nucleus of budding yeast.

3. LAS17-GFP does not show any nuclear localization, and extensive studies from my own laboratory (which are unfortunately not published) have used hypersensitive DAM-ID methods to detect LAS17 in the nucleus under various conditions of stress, but failed to detect any increase in response to damage. This, combined with the fact that LAS17 does not have a nuclear import nor export signal, makes it very unlikely that it is stabilizing an abundant factor like RPA at ss DNA regions in the nucleus.

It is clear that the authors cannot accept an argument based on unpublished data. However, they must at the very least, entertain the argument that the loss of las17 (the ts mutation las17-14 or Las17 degradation) acts in an indirect manner, affecting Mec1 activation and checkpoint response by altering a large number of protein-protein interactions.

Again in unpublished data, it was shown that LAS17 forms aggregates near the Golgi or ER on Zeocin, which induces single- and double-strand breaks, and abasic sites. Loss of las17 strongly affects survival of all types of stress, not only those that induce DNA damage. Thus the authors must at the very least entertain the fact that what they convincingly demonstrate for mammalian WASp is not conserved in yeast.

My hypothesis is that the replication fork / DNA damage repair defects in yeast stemming from the loss of LAS17 reflect alterations in G/F actin ratios. There is no doubt that loss of Las17 changes this, as it is the key activator of Arp2/3 and stimulates actin filament formation on its own (as an actin chaperone). However, in budding yeast there is no evidence for actin filaments in the nucleus (unlike those found in mammalian cells under stress). Instead, in yeast the function of actin dependent remodelers are compromised upon alteration of G/F actin balance, and this likely contributes to the

repair phenotypes observed.

The authors overinterpret their yeast data in manner aligned with their lymphocyte observations. I do not doubt what they measure in T- and B- cells, nor do I doubt the strong effect of *las17* depletion or mutation on repair in yeast. But if LAS17 protein were acting by stabilizing RPA then its deficiency should be at least partially compensated by RPA overexpression. This experiment did not yield the desired results.

In my opinion, without direct evidence (e.g. CHIP data for LAS17 at R-loops in yeast), their interpretation of Rad52 foci, R-loop increase and reagent sensitivity, can be explained by a variety of mechanisms that do not require the presence of Las17 in the cell nucleus. I think the yeast data should be omitted, but at the very least, their preferred interpretation must be softened by considering the fact that LAS17 has never been shown to be nuclear in budding yeast, and it lacks the necessary NLS for nuclear relocation.

Response: *We appreciate the discussion. The key point raised relates to whether scLas17, like human WASp, translocates to the nucleus or not? The premise for this argument is based on reviewer's observations :1) scLas17 does not contain a classical NLS (sNLS), and 2) their own unpublished work in which they are unable to demonstrate nuclear localization of scLas17.*

The above question emerges because proteins that shuttle between the cytosol and nucleus typically contain a transport sequence, both for entry into (NLS) and exit out of (NES) the nucleus through their association with the karyopherins via the classical importin pathway. However, certain proteins that do not carry a classical NLS (cNLS) are still able to translocate into the nucleus, some of them may contain a nonclassical NLS (ncNLS) or a cryptic NLS. As such STAT1, a transcription factor, does not contain the cNLS, but upon dimerization, a basic amino acid stretch is created that functions as a nuclear entry motif. SMAD proteins also do not contain the cNLS (PMID: 12048190; PMID: 34022911). In addition, post-translation modification (PTM), e.g., SUMOylation also functions to promote nuclear translocation by enabling association with karyopherins (Kaps) or nucleoporins (Nups). Accordingly, lack of cNLS alone does not preclude nuclear transport of a protein.

Nonetheless, we will state the following in the *Discussion*:

"It should be noted that since Las17 has not yet been shown in the nucleus of budding yeast, Las17 may influence these events directly or indirectly, the latter for example by altering the mono/oligomeric vs. polymeric actin balance in the nucleus."